# Influenza antibody breadth and effector functions are immune correlates from acquisition of pandemic infection of children

Janice Z. Jia [1,8], Carolyn A. Cohen[1,8], Haogao Gu [2,8], Milla R. McLean[3], Raghavan Varadarajan [4], Nisha Bhandari [4], Malik Peiris [2,5], Gabriel M. Leung [6,7], Leo L. M. Poon [1,2,5], Tim Tsang [6], Amy W. Chung [3], Benjamin J. Cowling [6], Nancy H. L. Leung [6] & Sophie A. Valkenburg [1,3] ✉

Cross-reactive antibodies with Fc receptor (FcR) effector functions may mitigate pandemic virus impact in the absence of neutralizing antibodies. In this exploratory study, we use serum from a randomized placebo-controlled trial of seasonal trivalent influenza vaccination in children (NCT00792051) conducted at the onset of the 2009 H1N1 pandemic (pH1N1) and monitored for infection. We found that seasonal vaccination increases pH1N1 specific antibodies and FcR effector functions. Furthermore, prospective baseline antibody profiles after seasonal vaccination, prior to pH1N1 infection, show that unvaccinated uninfected children have elevated ADCC effector function, FcγR3a and FcγR2a binding antibodies to multiple pH1N1 proteins, past seasonal and avian (H5, H7 and H9) strains. Whereas, children that became pH1N1 infected after seasonal vaccination have antibodies focussed to seasonal strains without FcR functions, and greater aggregated HA-specific profiles for IgM and IgG3. Modeling to predict infection susceptibility, ranked baseline hemagglutination antibody inhibition as the highest contributor to lack of pH1N1 infection, in combination with features that include pH1-IgG1, H1-stem responses and FcR binding to seasonal vaccine and pH1 proteins. Thus, seasonal vaccination can have benefits against pandemic influenza viruses, and some children already have broadly reactive antibodies with Fc potential without vaccination and may be considered 'elite influenza controllers'.

Influenza viruses cause seasonal epidemics and sporadic pandemics and, thus, are a significant threat to public health. Children are particularly susceptible to influenza virus infection and represent the highest number of hospital admissions during seasonal epidemics, whilst individuals with comorbidities and older adults have higher mortality. Seasonal influenza vaccines have limited protection against pandemic virus infection, yet in the 2009 H1N1 pandemic, seasonal vaccine efficacy from infection in children was 47% which is higher

[1]HKU-Pasteur Research Pole, School of Public Health, Li Ka Shing Faculty of Medicine, The University of Hong Kong, Hong Kong, SAR, China. [2]Division of Public Health Laboratory Sciences, School of Public Health, Li Ka Shing Faculty of Medicine, The University of Hong Kong, Hong Kong, SAR, China. [3]Department of Microbiology and Immunology, Peter Doherty Institute for Infection and Immunity, University of Melbourne, Melbourne, Australia. [4]Molecular Biophysics Unit, Indian Institute of Science, Bangalore, India. [5]Centre for Immunology and Infection (C2i), Hong Kong Science and Technology Park, Hong Kong, SAR, China. [6]WHO Collaborating Centre for Infectious Disease Epidemiology and Control, Li Ka Shing Faculty of Medicine, The University of Hong Kong, Hong Kong, SAR, China. [7]Laboratory of Data Discovery for Health Limited, Hong Kong Science and Technology Park, Hong Kong, SAR, China. [8]These authors contributed equally: Janice Z. Jia, Carolyn A. Cohen, Haogao Gu. ✉e-mail: sophie.v@unimelb.edu.au

than expected[1], whilst adults over 45 years of age also had higher than expected protection[2]. This may be due to cross-reactive antibody responses despite a lack of neutralization function providing some residual protection from infection and subsequent disease, which could be harnessed for universal vaccine development. Identification of immune correlates of protection is fundamental for next-generation vaccine design, and alternate approaches for assessing the contribution of non-neutralizing antibodies to limit infection are needed according to the framework for universal vaccine development[3].

Children need a higher level of HAI antibodies, estimated at 1:110 for 50% protection against infection[4], compared to the WHO accepted standard of 1:36 from adults for 50% protection[5], which only provides 22% protection in children. This has been attributed to additional immune correlates in adults, such as non-neutralizing antibody functions and T-cell responses, lowering the protective antibody threshold in adults due to the co-contribution of different immune factors. In a previous household study of pH1N1 infection, HA-stem-specific antibodies played an important role in protection[6] and may mediate their action through the Fc Receptor (FcR) function.

Some antibodies can cross-react between seasonal, pandemic, and avian influenza viruses and may have a protective role against limiting the acquisition or severity of influenza virus infection. Serum antibodies, typically IgG, can mediate effector functions such as antibody-dependent cellular cytotoxicity (ADCC), directing immune cells to kill infected cells or engulf them by antibody-dependent phagocytosis (ADCP), which are respectively mediated by antibody cross-linking of activating Fc gamma receptors (FcγR) 3a and FcγR2a engagement on Natural Killer (NK) cells and Macrophages. ADCC antibodies can be highly cross-reactive for different strains and even subtypes of influenza, whereby ADCC antibodies towards the pandemic H1N1 (pH1N1), highly pathogenic avian H7N9, and H5N1 influenza A viruses have been found from the blood of healthy unexposed adults[7]. There is an age-related accumulation of ADCC-mediating H7N9 specific antibodies, and individuals with ADCC cross-reactivity to group 1 HA (e.g., H5N1) also have high ADCC antibodies to group 2 (H7N9) viruses[8]. pH1N1 infection significantly boosted ADCC responses in children, which was less evident in adults[9]. Therefore, cross-reactive ADCC antibodies may target conserved viral regions of the HA, such as the HA-stem[10], or internal proteins, such as the Nucleoprotein (NP)[11] and Matrix 1 (M1), facilitating FcR-binding antibodies to have greater subtype cross-reactivity than conventional neutralizing antibodies[8,11]. Indeed, influenza A viruses can be phylogenetically distinguished based on HA groups 1 and 2, which have distinct HA-stems[12]. ADCP similarly has a high level of cross-reactivity but uses a different FcR, FcγR2a, expressed by B cells and Macrophages[13]. IgG3 antibodies typically have the highest affinity for FcγR3a and FcγR2a, but IgG3 has a relatively short half-life compared to the highly abundant IgG1, which has an estimated half-life that is 4 times longer. Therefore competition can occur between IgG1 and IgG3 responses for FcR receptors with time since exposure[14]. Whilst other antibody isotypes, IgA and IgM, bind to different FcR receptors and components at their tissue sites to mediate immune regulation and pathogen clearance[15–17].

In this study, we aimed to assess influenza-specific antibody breadth and function between sH1N1 vaccination and pH1N1 infection using archived samples from a randomized placebo control trial (RCT) of seasonal inactivated influenza vaccines in school-aged children that were collected at the onset of the 2009 H1N1 pandemic. We measured antibody responses by their isotype and subclass, FcγR3a and FcγR2a, binding against diverse HA, NA, or conserved NP and M1 proteins before and after seasonal vaccination, over time, and during the 2009 pandemic for acquisition of pH1N1 infection. We also explore features of antibodies that may correlate with reduced pH1N1 infection. This current study selected samples to address research gaps in antibody effector functions: (i) to determine vaccine-induced ADCC changes, (ii) longitudinal durability of vaccine-induced antibody FcR binding and isotype changes, (iii) differences in baseline FcR binding and isotype with the acquisition of pH1N1 infection, and (iv) the diversification of HA-specific antibody responses with vaccination and infection.

## Results

### Study background and sample selection

We leveraged an existing biobank of immune sera from a randomized vaccine placebo-controlled trial (RCT) established before and during the 2009 H1N1 pandemic to assess antibody breadth and effector functions as immune correlates from pandemic infection. The trial was established to assess seasonal influenza vaccine efficacy in school-aged children, 6–17 years old, initially from 119 households, including 71 households in which a child received influenza vaccination, in November 2008 in Hong Kong[18], and expanded in 2009 to 796 households, including 479 households in which a child received influenza vaccination and monitored for the following 6 years for infection outcomes (NCT00792051)[1]. Children were monitored for serologically confirmed acquisition of influenza infection (S1) by yearly post-epidemic/season sampling for standard haemagglutinin antibody inhibition (HAI) and infection was inferred by >4-fold rise in HAI response or inferred as uninfected children who remained seronegative (S0). Where influenza-like illness was reported, daily symptom diaries were recorded, and nasal swabs were collected for real-time polymerase chain reaction (RT-PCR) sampling to determine viral loads.

In this current study, participants were selected (Table 1) from this initial larger RCT study from various timepoints on the basis of seasonal vaccination (V1) or placebo (V0) and pH1N1 infection inferred by sero-conversion (S1) or no seroconversion (S0) against pH1N1 by HAI (>4 fold rise) within the first year of the study. There were 4 possible types of samples used, V1S1 (vaccinated infected), V1S0 (vaccinated uninfected), V0S1 (unvaccinated infected), and V0S0 (unvaccinated uninfected), at timepoints of either pre-vaccination (day 0), post-vaccination pre-infection (day 30), and post-infection (1 year and subsequent years). Children were selected for 4 different sub-studies for antibody studies (Table 1), (i) to determine vaccine-induced ADCC changes (Fig. 1), (ii) longitudinal stability of vaccine-induced antibody FcR binding and isotype changes (Fig. 2), (iii) differences baseline FcR binding and isotype with acquisition of pH1N1 infection (Figs. 3 and 4), and (iv) the diversification of HA-specific antibody responses with vaccination and infection (Fig. 5). The group size, age and gender were comparable between sub-studies, as uninfected controls were selected to match infected children. The majority of vaccinated children seroconverted (by HAI for >4 fold rise) to sH1 (55–90% of each sub-study V1 group) but not pH1 (0–17% of each sub-study V1 group), whilst fewer placebo children seroconverted to sH1 (0–12% of each sub-study V0 group) and pH1 (6–20% of each sub-study V0 group) (Table 1), consistent with previous reports of the larger cohort[1]. Whilst the frequency of participants that had high baseline (>40 HAI titer) (after vaccination, pre-infection) pH1 responses trended to be lower in V1S1 and V0S1 groups but were comparable for sH1 across the four groups (Table 1). Thus, differences in baseline HAI exist in the cohort that are related to infection outcomes and determining these in finer detail are the focus of this study.

### Cellular NK cell effector functions correlate with FcγR3a binding assays and are increased by seasonal vaccination

In sub-study (i), paired pre-post vaccination serum samples ($n = 30$ per group) (Fig. 1a) showed that seasonal TIV vaccination resulted in significant ($p < 0.0001$) rises in sH1N1 HAI responses in vaccinated children, but not towards pH1N1, or in unvaccinated children (V0) (Fig. 1b).

To assess pH1N1 HA-specific ADCC effector functions, CD107a degranulation by protein-specific antibody cross-linking was assessed using an NK cell line (Fig. 1c), which is stably transfected with FcγR3a. Seasonal vaccination (V1) increased NK cell ADCC responses for pN1 ($p = 0.008$) and pH1 ($p = 0.02$) (Fig. 1d) at day 30 from day 0 responses,

**Table 1 | Participant characteristics of sub-studies and HAI responses at placebo or vaccination timepoints**

| Sub study | Fig. 1 (i) Vaccination-induced ADCC changes | | Fig. 2 (ii) Longitudinal stability of vaccine antibodies | | Figs. 3 and 4 (iii) Protection from pdmH1 infection by seasonal vaccination status | | | | Fig. 5 (iv) Diversity of antibody responses elicited by seasonal vaccination or pandemic infection | | | |
|---|---|---|---|---|---|---|---|---|---|---|---|---|
| **Approach** | Cellular ADCC | | Cellular ADCC and multiplex | | Cellular ADCC and multiplex | | | | Multiplex | | | |
| **Group** | V1 | V0 | V1 | V0 | V1S1 | V1S0 | V0S1 | V0S0 | V1S1 | V1S0 | V0S1 | V0S0 |
| Sample size | 30 | 30 | 10 | 10 | 30 | 27 | 32 | 25 | 13, 7, 14 | 9, 9 | 7, 17 | 13 |
| Number of timepoints | 2 | 2 | 7 | 7 | 1 | 1 | 1 | 1 | 3 | 2 | 2 | 1 |
| Sampling timepoint | Pre (day 0) and post-vaxx (day 30) | | Pre (day 0), post vaxx (2009), 2010, 2011, 2012, 2013, 2014 | | Post-vaxx and pre-inf | | | | Day 0, day 30, 1 year | Day 0, day 30 | day 30, 1 year | Day 30 |
| Interval between day 30 and 1 year (months, mean ± SD) | – | | – | | – | | | | 10.78 ± 1.25 | – | 10.85 ± 1.5 | – |
| Age (years, mean ± SD) | 9.9 ± 2.7 | 10 ± 2.5 | 11.6 ± 1.8 | 10.6 ± 3.2 | 9 ± 2.6 | 9.8 ± 2.7 | 10 ± 2.4 | 10 ± 2.6 | 9.7 ± 2.6 | 9.6 ± 3.0 | 10.2 ± 2.4 | 9.6 ± 2.3 |
| Age (median) | 10 | 10 | 12 | 11.5 | 8.5 | 9 | 10 | 11 | 9 | 9.5 | 10.5 | 10 |
| Age range | 6–15 | 6–15 | 8–14 | 6–15 | 6–15 | 6–15 | 6–15 | 6–14 | 6–15 | 6–16 | 7–14 | 6–13 |
| Gender (% female) | 53 | 40 | 30 | 40 | 53 | 56 | 53 | 65 | 59 | 30 | 67 | 39 |
| Average GMT HAI rise sH1 (mean ± SD) | 106 ± 221 | 5 ± 16** | 185 ± 333 | 0.8 ± 0.4 | 49 ± 73 | 64 ± 121 | 4.5 ± 12.6 | 2.4 ± 6.2 | 39 ± 69 | 50 ± 55 | 2.6 ± 7.3 | 1.65 ± 1.9 |
| Freq of participants with >4-fold rise sH1 (%) | 67 | 7** | 90 | 0 | 70 | 55 | 9 | 12 | 69 | 80 | 5 | 8 |
| Average GMT HAI rise pH1 (mean ± SD) | 4.8 ± 12 | 2.5 ± 6 | 7.4 ± 20 | 3.4 ± 4.9 | 0.8 ± 0.7 | 3.6 ± 12 | 1.6 ± 2.7 | 2.1 ± 4 | 1 ± 0.8 | 0.8 ± 0.4 | 2 ± 3 | 3.2 ± 5.6 |
| Freq of participants with >4-fold rise pH1 (%) | 17 | 10 | 10 | 20 | 3 | 3 | 6 | 8 | 6 | 0 | 11 | 15 |
| Freq of participants baseline HAI+ sH1 (%) | 47 | 50 | 40 | 50 | 83 | 92 | 53 | 64 | 87 | 90 | 44 | 54 |
| Freq of participants baseline HAI+ pH1 (%) | 50 | 50 | 60 | 40 | 0 | 44 | 6 | 60 | 0 | 30 | 11 | 76 |

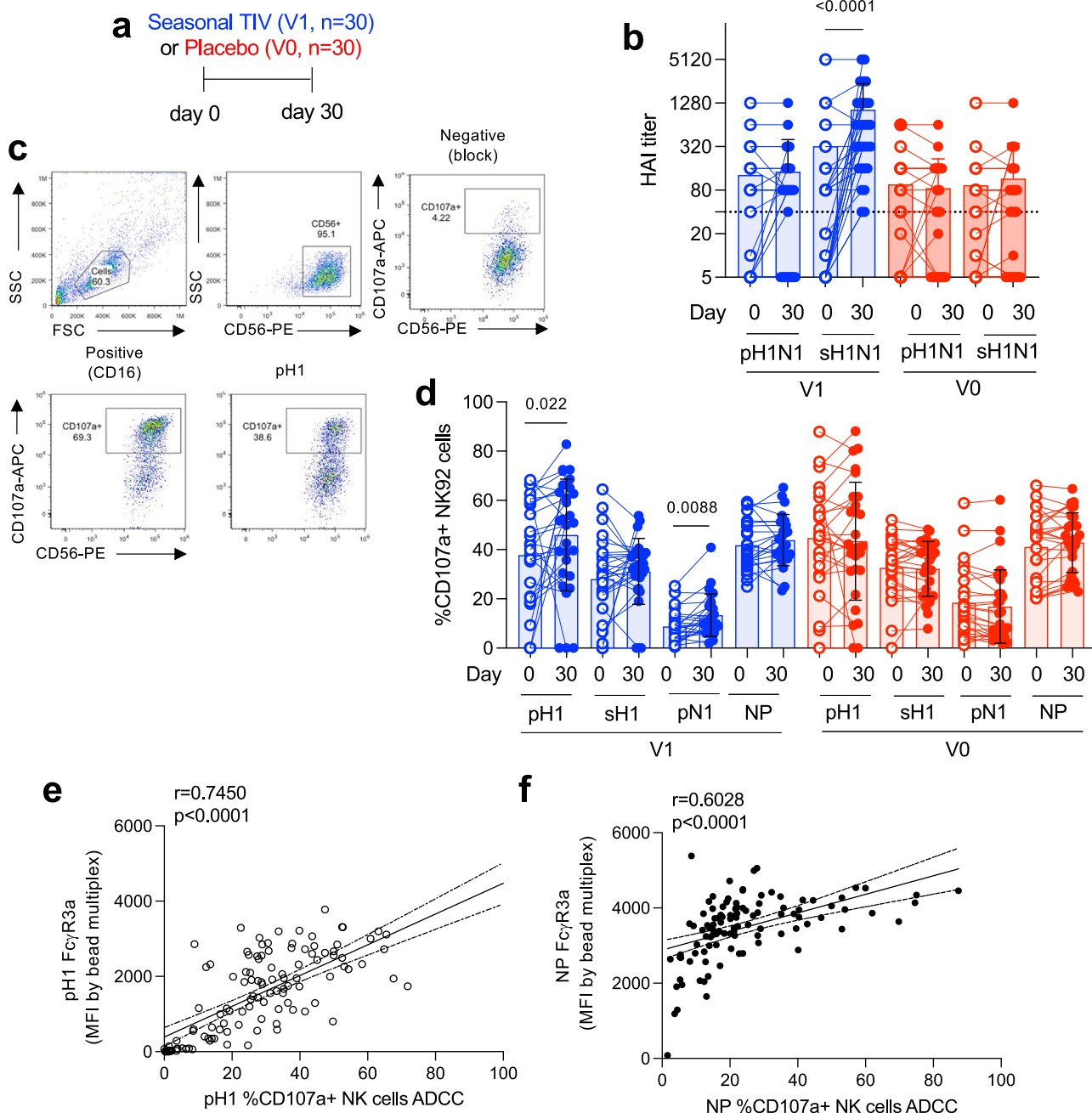

**Fig. 1 | Seasonal vaccination results in seasonal HAI-based seroconversion and pH1N1-specific ADCC.** Sub-study (i), **a** Children were vaccinated with seasonal Trivalent inactivated influenza vaccine (TIV, V1, *n* = 30) containing H1N1 A/Brisbane/2007 (sH1N1) or placebo (V0, *n* = 30), which assessed HAI titers **b** for sH1N1 and pandemic A/California/2009 (pH1N1). **c** Representative FACS plots of cellular function for ADCC activity by CD107a degranulation of NK92 cells for antibody binding to nonspecific protein (negative), CD16 (positive), or influenza proteins of interest (pH1). **b**, **d** Serum paired samples for pre (day 0) versus post (day 30) vaccination (V1) or placebo controls (V0) were tested in the functional NK92 cellular ADCC assay **d** for pH1, sH1, pN1, and NP proteins. Wilcoxon matched-pairs two-tailed signed rank test, pre versus post. Multiplex antibody binding assays (by FcγR3a binding to pandemic H1N1 HA protein (pH1) (**e**) and NP (**f**) from pooled groups of vaccinated and unvaccinated children (from Fig. 3) correlated with degranulation by CD107a+ of NK cell line. **b**, **d** Data presented as individual values, mean and SD, significance by two-tailed Wilcoxon test. **e**, **f** Lines represent trend by linear regression and 95% confidence interval. Significant correlations were measured using Spearman's correlation with the *r* and *p* values shown. Statistical significance is shown.

whilst there was no difference in functional ADCC responses between V1 and V0 groups at day 0. These vaccine-boosted responses that elicit NK cell degranulation strongly correlated with FcγR3a dimer binding for antibodies against pH1 (Fig. 1e, *r* = 0.7450, *p* < 0.0001) and NP proteins (Fig. 1f, *r* = 0.6028, *p* < 0.0001). Therefore, due to limited serum volumes collected nearly 15 years ago from a pediatric cohort and confirmation of the correlation between FcR dimer binding to NK cellular functional degranulation[13], multiplex assays using FcR dimers were used to evaluate multiple responses in further samples.

## Durability of antibody responses by vaccination
We aimed to measure antibody durability to determine if antibody responses were maintained long-term post-vaccination, with yearly sampling for 5 years. In sub-study (ii), to assess the durability (versus

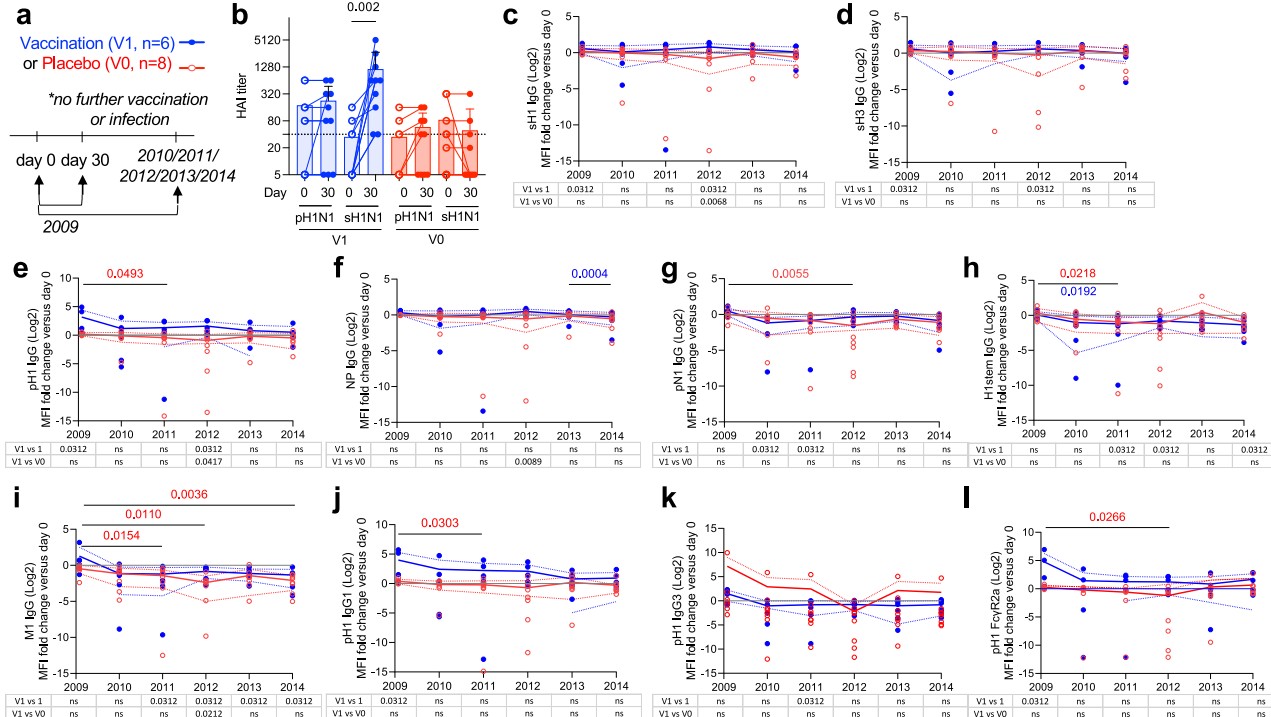

**Fig. 2 | Antibody durability post-vaccination.** Sub-study (ii), children were sampled pre (day 0) versus day 30 (2009) or yearly sampling over 5 years (2010–2014) for vaccinated (V1, *n* = 8) or placebo children (V0, *n* = 6) (**a**). Selected children had no further influenza vaccination or infection during this time. **b** HAI responses to sH1N1 and pH1N1 were assessed, data presented as individual values, mean and SD, significance by two-tailed Wilcoxon test. Antibodies were measured by multiplex bead assay for pre-vaccination responses (day 0) versus day 30 (labeled as 2009), 2010, 2011, 2012, 2013, and 2014 samples. **c–l** Data represents the fold change (from day 0 baseline) in mean fluorescent intensity (MFI). The experiment was repeated twice. Graphs are shown for proteins and detectors that had significant increases at 2 or more timepoints. Significant differences were identified for sH1-IgG (**c**), sH3 IgG

(**d**), pH1 IgG (**e**), NP IgG (**f**), pN1 IgG (**g**), H1stem IgG (**h**), M1 IgG (**i**), pH1 IgG1 (**j**), pH1 IgG3 (**k**), pH1 FcγR2a (**l**). Data represents individual values and line represent group means, and dotted lines represent SD. Statistically significant differences shown above (by a mixed linear model with Tukey's test correcting for multiple comparisons, between years with *p* values colored for comparison group V1 or V0, two-tailed) and in tables below versus V0 group (by a mixed linear model with Tukey's test correcting for multiple comparisons, two-tailed) or theoretical value of 1 (for no fold change, by one sample Wilcoxon test, two-tailed). Log-transformed *z*-scored fold-change data, including non-significant longitudinal responses, are shown in Supplementary Fig. 1 as a heat map.

day 0 baseline, day 30 (post-vaccination, 2009) at 1 year follow up in 2010, 2011, 2012, 2013, 2014) of seasonal vaccine stimulated antibody responses (IgG, IgG1, IgG2, IgG3, IgA, FcγR3a and FcγR2a as detectors) for seasonal and influenza proteins (H1 (sH1), pH1, sH3, H1-stem, pN1, sN1, M1 and NP) by multiplex approach. Children were also selected based on the criteria of no further vaccination or infection (by seroconversion) during the subsequent 5 years (Fig. 2a), resulting in a limited sample size of V1 (*n* = 8) and V0 (*n* = 6) due to stringent criteria, which precludes strong conclusions in antibody durability. We are unable to exclude the possibility of infection occurring between monitoring periods that was asymptomatic and resulted in less than a 4-fold HAI response; however, this remains unlikely based on baseline HAI titers and GMT rises (Table 1) and no changes in NP-specific antibody responses. Again, vaccinated children had significant rises in sH1N1 HAI responses (*p* = 0.002) but not pH1N1 virus (Fig. 2b).

Significant fold change increases (versus the theoretical value of 1) were observed shortly after vaccination (day 30, as 2009) for sH1-IgG, sH3-IgG, pH1-IgG, pH1-IgG1 and pH1-FcγR2a (Fig. 2b–l), but each returned to baseline within one year at 2010 (ns), and were not elevated compared to unvaccinated children. Other responses were analyzed but were not significantly increased by vaccination (Supplementary Fig. 1). Significant waning (by mixed effects model) of responses in unvaccinated (V0) children is evident across a number of antibody measures between 2009 to later years for pH1-IgG (Fig. 2e), pH1-IgG1 (Fig. 2j), pH1-FcγR2a (Fig. 2l), pN1-IgG (Fig. 2g), H1stem-IgG (Fig. 2h), and M1-IgG (Fig. 2i).

## Features of antibody responses prospective to pandemic infection

In sub-study (iii), to assess prospectively for antibody correlates prior to infection, we selected children (*n* = 25–30, Table 1) known seroconvert (S1) to pH1N1 in the next year or not (S0) and used their day 30 post-vaccination pre-infection timepoint serum samples (Fig. 3a). Children were initially randomized for vaccination, and uninfected children were selected to age and gender match infected children. Differences in HAI responses at the day 30 post-vaccination pre-infection timepoint were evident (Fig. 3b), with vaccination leading to significantly higher (*p* < 0.0001) sH1N1 HAI responses than pH1N1 responses. Furthermore, S1 children had significantly lower pH1N1 HAI responses than S0 children in both vaccinated and unvaccinated groups (*p* < 0.0001).

These prospective (baseline to infection) samples were assessed in an NK cellular function assay for ADCC responses against pH1N1 and seasonal vaccine representative proteins (Fig. 3). There was no difference in baseline sH1 ADCC responses between the 4 groups, vaccinated uninfected (V1S0), vaccinated infected (V1S1), unvaccinated (placebo) uninfected (V0S0) and unvaccinated infected (V0S1) (Fig. 3c). Whilst, significantly higher responses in V0S0 children compared to V0S1 were found for pH1 (*p* = 0.0373, Fig. 3d) and pN1 (*p* = 0.0034, Fig. 3e) ADCC antibodies. Furthermore, the NP-ADCC response in V0S0 children was also significantly higher than in V1S0 children (Fig. 3f, *p* = 0.0367).

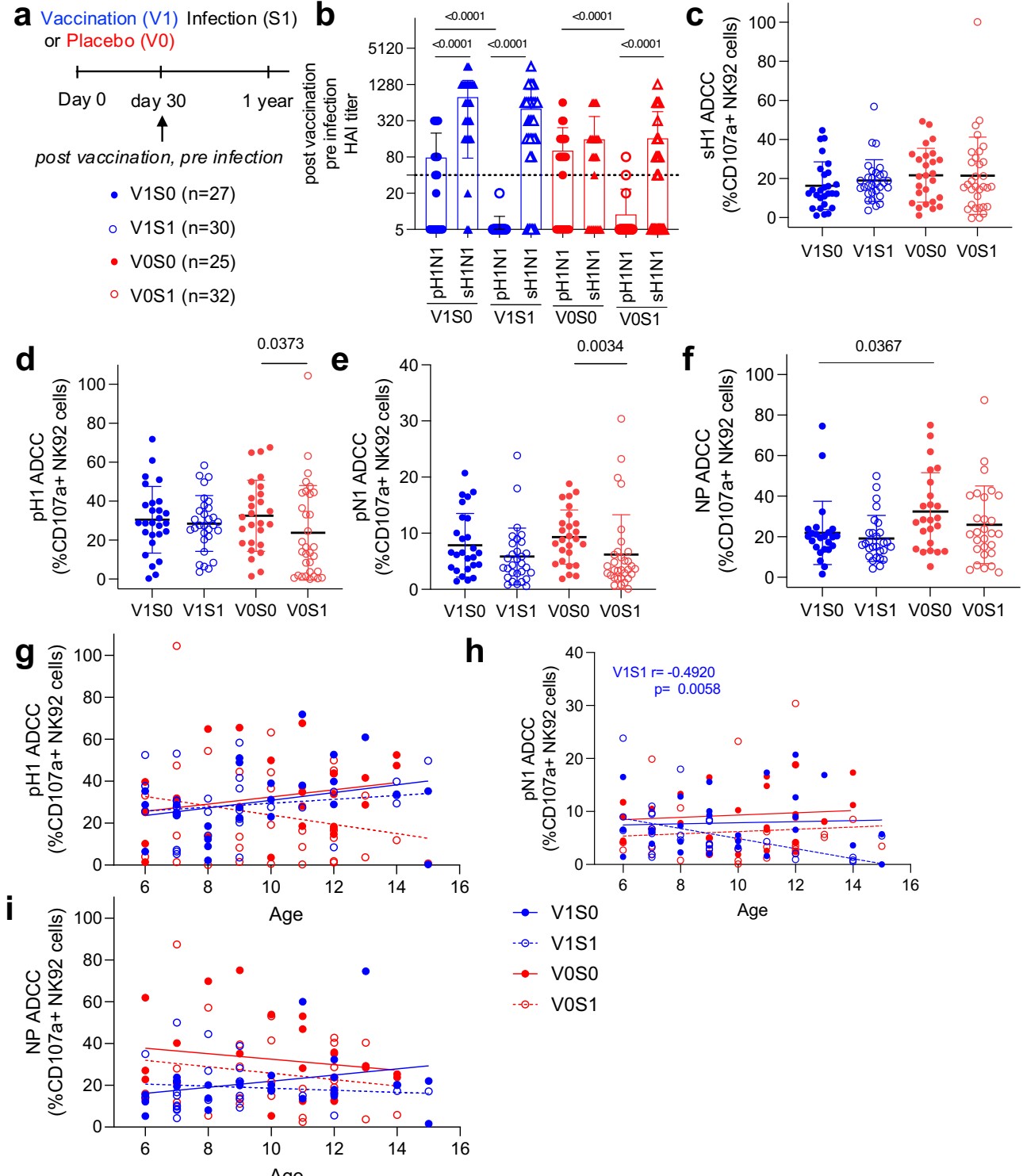

**Fig. 3 | Antibody function for NK cell activation with vaccination and pandemic infection.** Sub-study (iii), (**a**) Selected samples day 30 post-vaccination pre-infection sampling timepoint for vaccinated uninfected (V1S0, *n* = 27), vaccinated infected (V1S1, *n* = 30), unvaccinated uninfected (V0S0, *n* = 25) and unvaccinated infected (V0S1, *n* = 32), were assessed for pH1N1 and sH1N1 HAI (**b**) and magnitude of ADCC antibodies in a NK cellular assay for sH1 (**c**), pH1 (**d**), pN1 (**e**) and NP proteins (**f**). **b–f** Data presented as individual values, mean and SD. The correlation between ADCC responses and age in each vaccine group against (**g**) pH1, (**h**) pN1, and (**i**) NP proteins is shown as simple linear regression, with Spearman's correlation assessing statistical significance shown within graphs where significant. The experiment was repeated twice. Comparisons between groups were performed using Kruskall–Wallis with Dunns multiple comparisons test, and statistical significance as shown by *p* values above data.

However, there was no difference (by Chi-square test) in the sH1N1 baseline HAI positivity rate (a HAI titer of ≥1:40) of the groups between V1S0 and V0S0 groups (Table 1), at post-vaccination/pre-infection, with 92% of V1S0 children versus 64% of V0S0 children being HAI+ to sH1N1. Whilst the pH1N1 HAI+ rate was also similar, at 60% of V0S0 children versus 44% of V1S0 children. However, children who became pH1N1 infected (S1) groups had lower pH1N1 HAI+ rates at baseline, with 0% of the V1S1 being HAI+ for pH1N1 and 6% of the V0S1 group.

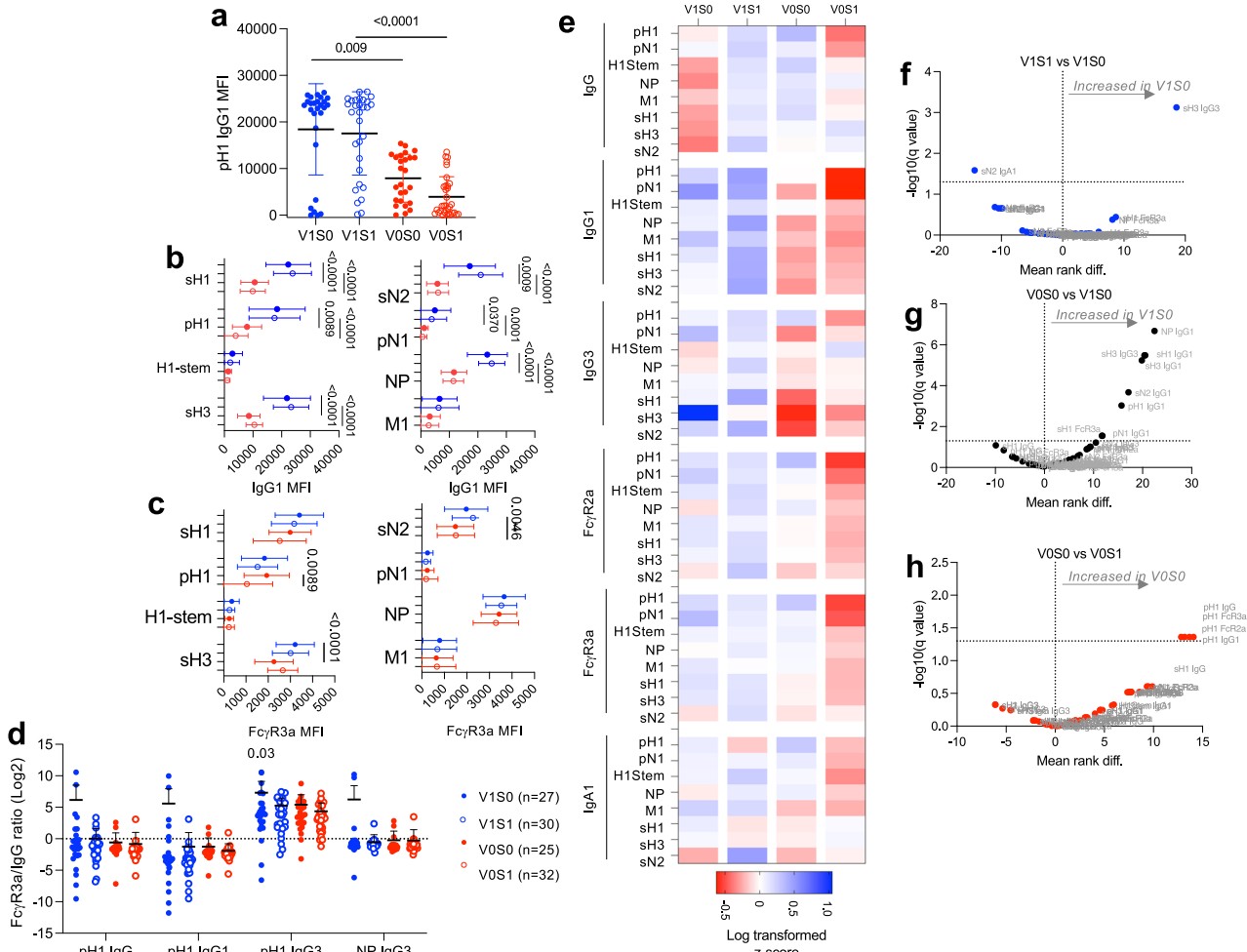

**Fig. 4 | Antibody features with vaccination and pandemic infection.** Sub-study (iii), Selected samples post vaccination pre-infection sampling timepoint (from **a**) for vaccinated uninfected (V1S0, *n* = 27), vaccinated infected (V1S1, *n* = 30), unvaccinated uninfected (V0S0, *n* = 25) and unvaccinated infected (V0S1, *n* = 32) (described Table 1), were assessed for antibody features by multiplex bead-based assay for pH1 IgG1 responses (**a**), data presented as individual values with mean and SD. Multiplex was performed for sH1, pH1, sH3, H1-stem, sN2, pN1, NP and M1 proteins for IgG1 (**b**), and FcγR3a (**c**) binding. Data presented as individual values, mean and SD. Experiment was repeated twice. Comparisons between groups were performed using Kruskal–Wallis with Dunn's multiple comparisons test and statistical significance as shown by *p* values above graphs. The ratio of MFI for FcγR3a and IgG responses from the same individual and response is shown (**d**), with a statistically significant difference from the theoretical value of 1 noted above by one sample *t*. Data for all antibody detectors are shown in heatmap representation (**e**). Experiment was performed in duplicate. The MFI of responses is shown as a heatmap (**e**), and the mean difference (FDR rate for multiple comparisons, *q*-value 1.3 as a dotted line for significant results by Mann–Whitney test) as a volcano plot analysis for V1S1 vs. V1S0 (**f**), V0S0 vs. V1S0 (**g**), and V0S0 vs. V0S1 (**h**).

There was no significant GMT rise in pH1N1 HAI responses or reported ILI in V0S0 or V1S0 groups from initial study recruitment, thus they were accounted for as pH1N1 infection naïve for sub-study (iii) and sub-study (iv) samples.

Correlation of age and ADCC antibody responses (Fig. 3f–h) showed an age-related increase in pN1 ADCC responses, with older 'vaccine failure' children (V1S1) having the lowest pN1 ADCC responses (Fig. 3g, *r* = −0.492, *p* = 0.0058), whereas no significant differences by exposure type were observed for pH1 (Fig. 3g) and NP (Fig. 3i) ADCC responses.

In the same sub-study (iii), the Multiplex assay was utilized to expand our analysis from the pre-exposure baseline to measure both binding (isotype/subclass) and functional binding antibodies against the pandemic, seasonal, and cross-reactive proteins (Fig. 4) for the same cohort of children and samples (as Fig. 3) (Table 1). Vaccinated children had significantly higher levels of IgG1 antibodies for the majority of seasonal vaccine and pandemic proteins (Fig. 4a and b), whereas no differences for IgG3 were detected (Fig. 4c). However, FcγR3a showed patterns related to infected S1 groups (Fig. 4d),

whereby V0S0 children had elevated pH1-FcγR3a responses compared to V0S1, and V1S0 children had elevated sH3-FcγR3a responses compared to V0S0, and sN2-FcγR3a responses were higher in V1S1 children compared to V0S1.

Whilst NP-IgG1 was elevated in vaccinated children (Fig. 4b), there was no difference in NP-specific FcγR3a binding (Fig. 4c), despite our earlier finding that V0S0 children had increased NP-specific ADCC cellular responses (Fig. 3e). Therefore, to assess the proportion of IgGs that may engage FcγR3a to mediate ADCC, we compared of the ratio of IgG of different subclasses (IgG, IgG1 and IgG3) to FcγR3a binding for pH1 and NP proteins (Fig. 4d). This showed an enrichment of pH1-FcγR3a to IgG3 detection, especially in the V1S0 group (*p* = 0.03) (Fig. 4d), thus seasonal vaccination likely increases pH1-specific IgG3 capable of binding FcγR3a. This is as expected as IgG3 has the highest avidity for FcγR3a to drive pH1-specific ADCC effector functions and is the first antibody developed upon protein antigen exposure (IgG3, IgG1, IgG2, then IgG4) (Fig. 4d).

A heatmap overview of log-transformed *z*-scored multiplex data highlights antibody features detected (Fig. 4e). Comparison of

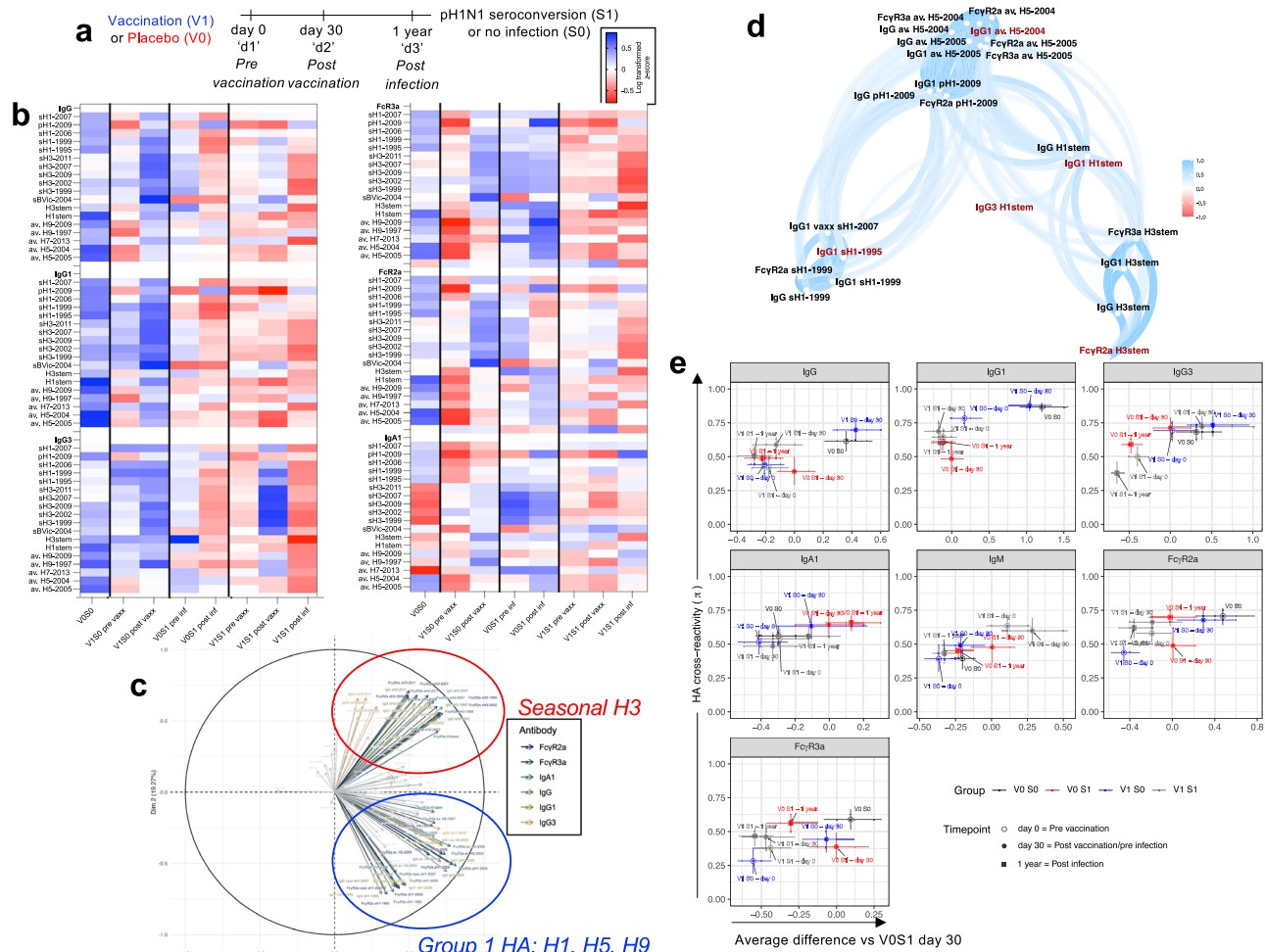

**Fig. 5 | HA antibody diversity with vaccination and pandemic infection.** Sub-study (iv) samples (**a**) for vaccinated uninfected (V1S0, $n = 9$ day 0, $n = 9$ day 30), vaccinated infected (V1S1, $n = 13$ day 0, $n = 7$ day 30, $n = 14$ 1 year), unvaccinated uninfected (V0S0, $n = 13$, day 30) and unvaccinated infected (V0S1, $n = 7$ day 30, $n = 17$ 1 year), were assessed for antibody features by multiplex bead-based assay for 19 different HA proteins of interest representing viruses from the seasonal vaccine, pandemic virus, related seasonal viruses and avian influenza viruses (Supplementary Table 1). The IgG, IgG1, IgG3, IgA1, FcγR2a, and FcγR3a response for each group and protein is shown (**b**). Data were log-transformed and z-scored prior to analysis. Experiment was performed in duplicate and twice for IgG. **c** Antibody profiles by principal component analysis combined from all groups at day 30 post-vaccination, pre-infection samples. **d** Correlation network analysis with selected variables (showing features selected by Elastic Net with frequency >80% in 2000 iterations are highlighted in red). Only variables with Pearson correlation coefficient >0.8 with at least one selected variable are shown, variables that are highly correlated are clustered together. The transparency of the path represents the strength of the correlation (less transparent, more solid = stronger correlation). **e** To measure $\pi$ HA cross reactivity antibody features, a 2-D representation of HA diversity ($\pi$) for individual antibody features for average MFI of HA proteins for the average difference versus unvaccinated infected subjects (V0S1) at day 30 (post-vaccination, pre-infection) timepoint samples (shown in red) were used as the basis of comparison for other groups. Data represents the mean (colored symbol) and 95% confidence interval (colored lines).

infected and uninfected vaccinated children showed that V1S0 children had significantly increased sH3-specific IgG3 and reduced sN2-IgA1 (Fig. 4f). Further, comparison of uninfected children for vaccinated versus unvaccinated showed vaccination significantly elevated IgG1 responses for NP ($q < 0.000001$), sH3 (and sH3-IgG3), sH1, sN2, pH1 and pN1 proteins, and sH1 FcγR3a the last parameter reaching significance ($q = 0.028$), with no increased pH1N1 FcR binding detected (Fig. 4g). Whilst uninfected unvaccinated children (V0S0) utilized pH1N1 reactive FcR effector functions for pH1 IgG, FcγR3a, IgG1 and FcγR2a-binding, which were all significantly increased ($q = 0.043$–$0.048$) compared to infected unvaccinated children (V0S1) (Fig. 4h).

These relative differences between antibody profiles of groups are reflected further in principal component analysis (PCA) (Supplementary Fig. 2), V0 children have a greater correlation of IgGs and FcRs in the same direction, whereas V1 children had an obvious separation

between IgG and FcR at the y-axis dimension. Vaccine-driven antibody responses may not effectively acquire FcR functions, and thus the immune responses that contribute to lack of infection in V1S0 versus V0S0 children may be different.

In a separate set of children's samples in sub-study (iv) (Fig. 5a, Table 1), including samples pre/post-vaccination, and pre/post-infection, we increased our protein panel to 19 seasonal and avian influenza HA proteins (Supplementary table 1) to determine antibody binding breadth (for IgG, IgG1, IgG3, IgA1, IgM, FcγR3a and FcγR2a binding). Seasonal HA proteins were selected based on strains circulating during the children's lifetime. Heatmap analysis of log-transformed z-scored data (Fig. 5b) showed a trend for S0 children to have higher responses than S1 children, especially for IgG, IgG1, and IgG3, which were increased by vaccination. V1S1 children tended to have lower responses across most proteins and antibody features, excluding post-vaccination IgG3 responses to seasonal H1 and H3 HA proteins, which

were not maintained post-infection, whilst elevated pH1-IgA1 was prominent. This was also reflected in changes in the PCA (Supplementary Fig. 3).

To break down antibody specificities for significant differences from the heat map, the mean difference between groups and detectors of interest was visualized by volcano plot analysis (Supplementary Fig. 4a–c). Vaccinated infected children, V1S1, had minimal differences to V1S0 children for IgG1 responses, with only seas. H1-1999 being significantly elevated ($q = 0.04$) (Supplementary Fig. 4a). Whilst naturally protected children (V0S0) compared to unvaccinated infected children (V0S1), had significantly reduced IgA1 to seasonal H3-HA's ($q = 0.042$, Supplementary Fig. 4b), and increased IgG1 to group 1 HAs of seasonal H1 ($q = 0.001$ and $0.002$), pH1($q = 0.01$) and avian H5 HA's ($q = 0.01$ and $0.03$) (Supplementary Fig. 4c). As many permutations of comparisons are possible, PCA was used to gather an overview of children's groups. A combined PCA of post-vaccination pre-infection samples, i.e., day 30 timepoint (Fig. 4c), which captured the total variance of the data (Dim. 1: 24% and Dim. 2: 18%), showed that H1-HA, influenza B, and avian (H5 and H9) antibody responses PC was not likely to be correlated to H3-HA responses. Correlation network analysis with Feature selection by Elastic Net of selected variables (Fig. 5d, Supplementary Fig. 4e) showed pH1 antibody features clustered with avian H5-specific IgG and FcγR2a and FcγR3a, separate from seasonal H1-HA's or HA-stem responses. Red labels correspond to the important variables selected by elastic net with frequency >70% of the total 2000 iterations as the important variables in terms of predicting infection.

From our heatmap and PCA analysis, disparities were apparent in the breadth of HA-specific antibody responses by vaccination and infection between groups. Therefore, a 2-D representation was created of responses of different groups and timepoints to compress for HA cross-reactivity (measured by π) versus V0S1 pre-infection samples (day 30 post-vaccination, pre-infection timepoint, selected as the most susceptible group to infection) (Fig. 5e). The *y*-axis represents the diversity of the HA response, and the *x*-axis represents the magnitude difference from V0S1 day 30 of other timepoints or groups responses. For the detectors of IgG1, FcγR2a, and FcγR3a, the V0S0 group had the most diverse antibody response with the highest cross-reactivity score (π) and a response magnitude that was significantly above V0S1 pre-infection responses. The V1S0 post-vaccination samples showed similar patterns to V0S0, with the additional feature of significantly increased IgA1 magnitude. Whereas breakthrough infections, i.e., V1S1 children, had the largest IgM and IgG3 responses of the study, and the magnitude and diversity of these responses were increased by vaccination (day 0 versus day 30) and were reduced following infection (1 year). Whilst V0S1 children at post-infection (1 year) acquired increased FcγR2a to a greater extent than FcγR3a responses compared to their pre-infection (day 30) responses, which may provide future breadth of immunity.

### Predictive antibody features as immune correlates against pandemic infection

To assess antibody features that protected from acquisition of pH1N1 infection, logistic regression models with stepwise variable selection by AIC were built for shared data from sub-study (iii) (Fig. 4) and sub-study (iv) (Fig. 5) samples, which included 4 proteins (i.e., sH1-2007, pH1, sH3-2007 and H1-stem) and 6 antibody features (i.e., IgG, IgG1, IgG3, IgA1, FcγR3a, and FcγR2a) (Supplementary Table 1), resulting in the comparison of 24 variables. To avoid overfitting and overparameterization, we input the top 10 most important variables shared for the sub-study (iii) and sub-study (iv) data by Elastic Net with PLSDA (determined from Supplementary Fig. 5d, see the "Methods" section). All input antibody responses are at post-vaccination pre-infection timepoints the output value is whether they become infected (S1 or S0 group), and we trained the model with sub-study (iii) data ($n = 115$ samples) and tested the model with sub-study (iv) data

($n = 36$ samples, day 30 post-vaccination, pre-infection timepoint) where there is no sample overlap. Three models were built, Model 1: antibody features (Supplementary Fig. 5a); Model 2: antibody features and HAI status to sH1 and pH1 (Supplementary Fig. 5b); or Model 3: HAI status to sH1 and pH1 only. Model 1 (Supplementary Fig. 5a) showed that sH3-2007-IgG1 and IgG3, followed by vaxx sH1-2007-FcγR3a, pH1-FcγR3a, H1stem-IgG, pH1-IgG1 and IgG3 as antibody features were significant to predict pH1N1 infection, whilst Model 2 ranked HAI titers to pH1 and sH3-2007-IgG1 first, followed by H1Stem-IgG1 and sH3-2007-IgG3. (Supplementary Fig. 5b). Three antibody features were selected in both sub-study (iii) (Fig. 4) and sub-study (iv) (Fig. 5) data by Elastic Net and prediction using PLSDA (Supplementary Fig. 5d), i.e., H1stem-IgG1, sH3-2007-IgG1 and sH3-2007-IgG3, which were selected in >50% replicates in both data, whilst FcR binding was not prominent. When using sub-study (iii) data as the training set (from Fig. 4), Model 1 had 75% accuracy on sub-study (iii) data and 36% accuracy on sub-study (iv) data (Supplementary Fig. 5c). Whilst for Model 2, adding HAI status slightly improved the accuracy (76%), and specificity (94%), It is notable that model 2 achieved similar fitting accuracy by using only four variables (selected by AIC), compared to seven variables in model 1. Both models had no sensitivity against sub-study (iv) sample data (Supplementary Fig. 5c). Thus, the performance of prediction is not high, however, input of more shared antibody features may improve the model. As a reference, model 3 using only HAI data can attain similar performance (Supplementary Fig. 5c); therefore, additional antibody features may contribute to protection from infection, or a larger sample size is needed to outperform traditional HAI measures.

## Discussion

Correlates of protection aid vaccine development, vaccine reformulation, clinical trial endpoints, and next-generation vaccine design. Seasonal influenza vaccination has variable vaccine efficacy depending on the age group, vaccine match, and year, with a reported efficacy of 10–60%[19]. Current influenza vaccines are under scrutiny for improved protection and there has been a call for the development of universal influenza vaccines to increase coverage for coverage to pandemic viruses. Non-neutralizing antibody functions can be highly cross-reactive between different influenza strains and subtypes and could be used as a target for broadly reactive vaccines that reduce the incidence and severity of pandemic infection. Assays to measure other antibody functions beyond HAI are needed for next-generation vaccines whose protection is not HAI-dependent. Seasonal influenza vaccination provided 47% vaccine efficacy from H1N1 pandemic infection in children[1]. Our study utilized an archived biobank from a randomized control trial as a prospective 2009 pandemic 'time capsule' to determine serological correlates of protection from pandemic infection; thus our data assess prospectively for correlates prior to infection among cohorts that would seroconvert to pH1N1 in the next year.

While schools were closed in April 2009 for two months to slow the spread of H1N1, ultimately, when schools restarted in September 2009, it was estimated that more than 50% of children were infected within a few months, and this is the period covered by our study. Monovalent pandemic H1N1 vaccine became available in January 2010 with a very low community uptake[1,20]. Therefore, this study aimed to determine the contribution of prospective serological responses that were associated with the lack of pandemic H1N1 infection.

Immune correlates can contribute to reduced acquisition of infection, viral shedding for transmission, or severity of infection (reviewed in ref. 21). Different immune variables will contribute at each level to a different but not exclusive extent, with mucosal antibodies providing a barrier defense or reduced onwards viral shedding (reviewed in ref. 22) and HAI antibodies contributing to blocking infection (reviewed in ref. 23). Whilst cross-reactive memory T cells are reported to contribute to protection from symptomatic infection[24], and can be boosted in children by live attenuated influenza vaccines[25].

The recruitment kinetics of T, B, and NK cells determining patient outcomes and can be critical for survival from novel subtypes[26]. As our study used samples that were from community-acquired pH1N1 infection and no reported hospitalization, our study is of immune correlates of protection from acquisition of infection, not disease severity.

Antibody waning is a natural feature of plasmablast B cell response contractions to form a stable B cell memory pool, and the half-life of different antibody subclasses varies substantially. Previous exposure to seasonal H1N1 viruses in older adults was attributed to reduced severity of pH1N1[2], and there is an age-related accumulation of ADCC responses[8]. Seasonal vaccination increases Fc effector functions of pH1N1 specific HA, NA, and NP antibodies; however, this appears to be short term and wanes within 1 year post-vaccination; however, greater waning rates were seen in unvaccinated children. Seasonal vaccination did not bolster FcR effector functions to corresponding seasonal-specific antibody responses, which may be due to maximized FcR functions from previous encounters with related seasonal H1N1 strains. Whilst gains in pandemic-reactive FcR function maybe attributable to vaccination briefly increasing different B cell sources. These features could be deciphered at the clonal level if cells were available.

Unvaccinated uninfected (V0S0) children also had increased FcR-mediated effector functions of pandemic-specific HA, NA, and NP antibodies. Furthermore, in V0S0 children, increased antibodies for NK cell function were consistent with increased pH1-specific IgG1, FcγR2a, and FcγR3a binding and further translated to these responses having increased HA-diversity compared to other groups. pH1 antibody features clustered separately to H3-HA responses (by PCA) and were related to cross-reactive avian H5-specific IgG, IgG1, FcγR2a, and FcγR3a responses (by correlation network analysis), suggesting these cross-reactive responses are less focussed and not trained by seasonal virus exposure of other groups. Particular immune or genotype features of V0S0 children should be further studied, similar to studies of HIV elite controllers, that may enable their immune capacity to pre-empt viral diversity.

Considering the 2D representation of responses, the magnitude and diversity of IgG1, FcγR2a, and FcγR3a responses were highest in V0S0 children than in other groups suggesting features of their antibody profile are more cross-reactive and with effector functions. Antibody responses in V1S0 children post-vaccination attained some of these functions and levels but not to the same extent as V0S0 children; however, they also had the highest IgA1 responses, which may contribute to their lack of pH1N1 infection. Whilst V1S1 children at post-vaccination pre-infection timepoint samples had elevated IgG3 and IgM responses, which compete with IgG1 for Fc receptors, and IgM points to a less differentiated response with lower class switching. Whilst IgG3 and IgM in V1S1 children decreased post-infection, there was not an accompanying increase in HA-diverse IgG1 or FcγR2a and FcγR3a binding responses but rather a reduction, suggesting that V1S1 children did not become V0S0-like in their responses post-infection and remain unable to fully maximize antibody effector functions. Thus, vaccination and prior infection cannot account for the lack of infection in V0S0 children or susceptibility of V1S1 children, and further host factors (reviewed in ref. 27) such as genotype for key immune pathways (e.g., IFTIM3, CD55, IL28B) or antigenic gaps in prior immunity (no previous or recent H1N1 exposure) should be investigated[28].

The logistic regression model and volcano plots of heatmap analysis showed that group 2 H3 HA-specific IgG3 antibodies were a negative predictor of infection, whilst seasonal H1 and pH1-IgG3 antibodies at pre-infection timepoints were positively associated, thus protective from infection. Similarly, other studies have also seen negative impacts of other elevated antibodies to other influenza subtypes, i.e., a New Zealand study for symptomatic influenza infection found elevated influenza B Yamagata responses[29] were associated with an increased risk of symptomatic influenza A infection. Whilst, in a Nicaraguan household study using protein microarrays, most children under 6 years of age had narrow HA-specific antibody responses, some children showed back boosting of H3N2-specific antibodies by pH1N1 infection, and pre-existing antibody breadth was associated with protection[30]. Whilst the combined logistic regression model had non-significant results for FcR contributions, results from the V0S0 group throughout point to the importance of elevated pH1N1-reactive IgG1, FcγR2a, and FcγR3a in this group as protective from pH1N1 infection that may extend to other avian H5 viruses based on antibody correlation networks.

Our study has some limitations, mainly that PBMC isolation was not performed; thus, no cellular assays were performed for participant NK cell or macrophage function, nor other immune correlates such as memory T and B cell responses, thus our data is all based on serological functions and binding to recombinant proteins. Furthermore, due to limited archival sample volume, we were unable to explore the role of glycans on the Fc region on specific antibodies that can modulate FcR interactions[31]. The prediction model for antibody functions could be improved by greater sample size and overlap of parameters between the study groups, as set 1 (Fig. 3) measured 56 antibody parameters and set 2 (Figs. 4 and 5) measured 133 antibody parameters, there were only 24 overlapping proteins and antibody features tested. Nevertheless, as our main outcome was pH1N1 infection, relevant parameters were included in both aims, but these antibody features did not surpass the contribution and accuracy of baseline HAI infection for pH1N1 infection susceptibility. However, as year 1 post-vaccination/placebo pH1N1 HAI seroconversion and ILI reporting was the selecting factor of our samples as S1 or S0 groups, it is not surprising that from children that can seroconvert, their HAI baseline titers are major predictors of infection. Further studies of participants that do not seroconvert, i.e., "vaccine or infection failures"[32], are needed to determine immunity in these groups that are not HAI dependent. On the other hand, while the selection of infected and uninfected individuals was matched by age and sex, comparison of their pre-infection antibody responses was conducted through prediction models based on machine learning[33] and only included different antibody responses as input variables; thus, interpretation of the importance of the antibody responses identified to be predictive in the present study should be cautious due to potential confounding by other factors (such as social contact pattern) that were unadjusted for in our models.

In HIV[34] and SARS-CoV-2 studies (reviewed in[35]), FcR effector functions are longer lived than neutralizing antibodies and can have therapeutic importance by modulating disease severity and viral loads. Furthermore, in HIV infection, ADCC mediating antibodies have been attributed to being moderately protective in the absence of broadly neutralizing Abs[36] and are correlated with ADCC and IgG3[34,37]. Furthermore, high FcR responses can be enriched in pediatric HIV non-progressors[38]. Whilst vaccine antibody breadth is associated with infection-free time from SARS-CoV-2 variants[39], similar to pre-existing antibodies pandemic and avian influenza found here. Therefore, FcR effector functions and antibody breadth are attractive features for immune correlates of universal vaccine design.

## Methods
### Study participants and selection for serum panels
To assess antibody features from seasonal vaccination that may contribute to reduced or delayed acquisition of pandemic H1N1 infection in our exploratory study, we selected archived samples (Table 1) from a randomized placebo-controlled trial (NCT00792051) and the subsequent follow-up previously conducted by our group in school-aged children (6–17 years of age) which reported Hemagglutinin antibody inhibition (HAI) titers[1,18]. In the original trial, which aimed to study the direct[1] and indirect benefits[40] of vaccinating children in households,

households with at least one child aged 6–17 years were enrolled, and one eligible child from each household was randomly assigned to receive either a seasonal trivalent inactivated influenza vaccine (TIV) or placebo at enrollment. Participant gender was comparable across groups (Table 1). The 2008–2009 and 2009–2010 TIV used in our study included the influenza A/Brisbane/59/2007(H1N1)-like, A/Brisbane/10/2007(H3N2)-like, and B/Brisbane/60/2008-like virus (VaxiGrip, Sanofi Pasteur), i.e. it included a seasonal H1N1 (sH1N1) virus strain that circulated before and was antigenically different from the H1N1 virus (pH1N1) that caused the 2009 influenza pandemic. Sera were collected from both children receiving the randomized study vaccination and their household members. In Year 1, most serum was collected from study subjects at baseline (day 0) before vaccination (August 2009 through February 2010) ("Pre vaxx"), 1 month after vaccination (day 30) ("Post vaxx", also referred to as "Pre inf"), and at the end of the follow-up period as postseason sampling (August to December 2010). Participants were subsequently followed for 5 years until the end of 2014 (Years 2–5), with the collection of (post-epidemic) sera and other information, including self-reported vaccination status between October to December each year in all participants ("Post epidemic"). Serologically confirmed infection in Year 1 is defined as a ≥4-fold GMT rise in HAI titer between day 30 and post-season sampling, and in Years 2–5 (2010-2014), as ≥4-fold GMT rise in HAI titer between two consecutive post-epidemic sera. Virologically confirmed infection is defined as PCR-positive in respiratory swabs collected during acute illness.

For the present study, we selected subsets of children 6-17 years of age who had received any influenza vaccination in Year 1 ("vaccinated"/"V1") or not ("unvaccinated"/"V0") for secondary analyses to study the effect of vaccination on short-term and long-term serologic responses in children who had ("infected"/"S1") or had not ("uninfected"/"S0") pH1N1 infection in Year 1 (Table 1). In sub-study (i), to study the effect of vaccination on cellular ADCC, we randomly selected 10 children from three age groups (≤8 years, 9-11 years, ≥12 years old) in both vaccinated and unvaccinated groups and tested their pre- and post-vaccination sera. In sub-study (ii), to study the longitudinal stability of vaccine antibodies, separately, we randomly selected 10 children each from vaccinated and unvaccinated groups, among children who did not have any influenza vaccination, nor any virologically or serologically confirmed infection by influenza A/sH1N1, A/pH1N1, A/H3, and B viruses, in subsequent years of follow-up until Year 5, and tested their pre-/ post-vaccination and five post-epidemic sera. In sub-study (iii), to study the protection against pH1N1 infection by vaccination, due to limited sample size, we selected all children with serologically confirmed pH1N1 infection in year 1 whether they were vaccinated or not, and randomly selected 10 children without serologically confirmed pH1N1 infection in year 1 from each of the three age groups in both vaccinated and unvaccinated groups, and tested their Pre inf (equivalent to Post vaxx) sera. Lastly, in sub-study (iv), to study the diversity of antibody responses after a combination of vaccination and infection, we randomly selected 5 children from each of the three age groups in both vaccinated and unvaccinated groups, in both uninfected and infected children with serologically confirmed pH1N1 infection in year 1. Overall, we studied the vaccine-induced short-term cellular ADCC, long-term cellular ADCC, IgG, IgA, and FcγR against multiple viral proteins, homologous (sH1N1) and cross-reactive (pH1N1) antibody response, cross-reactive response (IgG, IgA and FcγR) against infection by pH1N1, and breadth (IgG, IgM, IgA, and FcγR) of response against multiple H1 and H3 virus strains by vaccination or infection (Supplementary Table 1).

Proxy written informed consent was obtained for all participants from one parent or legal guardians, with additional written assent from those ≥8 years of age. The study protocol was approved by the Institutional Review Board of Hong Kong University (UW 08-008).

## FcR protein expression and biotinylation

pCR3 plasmids encoding for dimeric high-affinity FcγR2a-H131 and FcγR3a-V158 with a C-terminal hexahistidine (His)-tag, and Avitag were kindly provided by Bruce Wines and Mark Hogarth (Burnet Institute, Australia). FcR dimer proteins were expressed in Expi293F cells and purified by TALON chromatography. Purified Avi-tagged proteins were biotinylated using a BirA 5000 biotin-protein ligase kit (Avidity) according to the manufacturer's instructions, and the reaction was performed overnight at 4 °C.

## Multiplex bead array for antibody specificity and FcR binding

To study antibody binding, we adopted a customized multiplex assay, adapted from Mclean et al.[13]. This assay has the capacity to simultaneously detect antibodies that react to multiple proteins in a single well for a single antibody detector. For each study aim, a panel of influenza proteins (Supplementary Table 2) was individually conjugated to beads of different regions as below and measured for different antibody detectors as listed. Recombinant full-length monomer-form HA, NA, and NP proteins (Supplementary Table 2) were purchased from Sinobiological, MyBiosource, Sigma Aldrich, Merck, and BEI resources. Custom headless HA-trimer ministem proteins for HA-stem proteins (from Raghavan Varadajaran, Indian Institute of Science)[41].

For the preparation of 1000 reactions, 100 µl of Bioplex non-magnetic carboxylated beads (BioRad) were centrifuged at maximal speed for 3 min and resuspended in 80 µl bead activation buffer (0.1 M $NaH_2PO_4$ in ddH2O, pH 6.2), and activated with 10 µl Sulfo_NHS (N-hydroxysulfosuccinimide, 50 mg/ml, ThermoScientific) and 10 µl EDC (1-Ethyl-3-(3-Dimethylaminopropyl) carbodimide, Hydrochloride, 50 mg/ml, Invitrogen) within 10 min. After incubation for 30 min on a rotator at room temperature (RT), beads were washed with coupling buffer (0.05 M MES in ddH2O, pH 5.0) twice and resuspended with coupling buffer and 0.05 mM of protein of interest to the activated beads for 2–3 h, (RT), with the final volume being up to 250 µl with coupling buffer. Beads were washed with 500 µl assay buffer (0.1% BSA in PBS, pH 7.4) and blocked with 500 µl blocking buffer (0.1% BSA, 0.02% TWEEN-20, 0.05% Azide in PBS, pH 7.4) for 30 min on a rotator. Then beads were washed with 500 µl of storage buffer (0.05% sodium azide in PBS), and the beads were resuspended in 125 µl of storage buffer. All assays were run on the same conjugation round within 1 month.

For the detection of protein-specific antibodies, coupled beads of different bead regions were mixed to a final concentration of 20 beads/µl for each kind of protein of interest in the assay buffer. A mixture of 1:1 beads was incubated with a 1:100 diluted plasma sample in duplicate on a plate shaker overnight. Beads were washed and incubated with anti-human detector-PE antibody or biotinylated FcγR for 2 h. For the detection of biotinylated FcγR, samples were incubated for an additional 30 min with 1 µg/ml streptavidin-PE (Biolegend). Samples were then washed with 150 µl assay buffer twice and resuspended in 100 µl of sheath fluid for detection. Multiplex bead assays were acquired using the MAGPIX system (Luminex, array reader v2.6.1, microplate platform v2.1.15, Bio-Plex manager software v6.2.0.175 from BioRad) following the manufacturer's instruction to detect the microspheres and binding of PE fluorescence to calculate a median fluorescence intensity (MFI). Background signal, defined as the average MFI observed for each microsphere set when incubated with the PE-conjugated detection reagent in the absence of a plasma sample, was subtracted from the MFI for each sample. A pool of 10 blood donors' sera was used to normalize data across plates as an internal control between assay runs, and total IgG and tetanus protein as positive control per sample per detector. Serum samples were run in duplicate. All assays were run on the same conjugation round within 1 month, with no degradation evident due to covalent linkages of proteins and beads. Protein conjugation efficiency was confirmed by

anti-His-PE detection[42], and the MFI was confirmed by pooled buffy pack serum in the expected range. Proteins that had both lower than 40% coupling efficiency and 20% MFI relative to pH1 across a range of detectors were excluded (data from one protein HA B/Vic-2008 was excluded). Due to differences in protein quality, and conjugation efficiency, no direct comparisons are made between protein-specific responses, and data is log-transformed then z-scored for heat map, PCA, and combined data analysis.

## Protein-specific NK cell-mediated ADCC functional responses

A protein plate-bound NK ADCC assay was adopted from Jegaskanda et al.[7]. The assay measures degranulation (CD107a) human NK cells (NK-92-FcγR3a-bearing cell line, provided by Fox Chase Cancer Center) due to crosslinking of influenza-specific IgG in donor sera. Briefly, U-bottom ELISA plates (Nunc-Immuno MaxiSorp, NUNC) were coated overnight at 4 °C with recombinant influenza proteins (Supplementary Table 2). Cells were verified as NK cells by CD56 staining for FACS analysis.

Background ADCC activity by NK cells was determined by paired serum with plate-bound non-specific protein (FBS block). Positive controls include purified CD16 (1:100, clone: 3G8, Biolegend) for coating and pooled human sera (n = 10 donors pooled) tested against each protein included in each experiment as internal controls. Responses were normalized to % of maximum CD107a+ of CD16 stimulated positive controls, and the background of negative controls was subtracted from each sample.

Serum was heat inactivated for 30 min at 56 °C and was typically tested at 1:20 dilution. Protein-bound plates were thoroughly washed with PBS and heat-inactivated immune plasma bound for 2 h at 37 °C. Plates were then further washed and incubated with NK92 cells ($2 \times 10^5$ cells per well). After 5 h, cells were stained with anti-human CD107a-APC (1:50, clone: H4A3) and CD56-PE (1:200, clone: 5.1H11) (both Biolegend) in FACs buffer (PBS with 1% FBS and 0.1% sodium azide). Cells were then fixed (BD Cytoperm/cytofix buffer), washed, acquired by Flow Cytometry (Attune NxT, Invitrogen), and analyzed by FlowJo software (v10).

## Statistical analysis

Statistical analysis was performed using GraphPad Prism v9 and Excel 16.7 software. To assess vaccine immunogenicity, $\chi^2$ tests were used to compare the proportions of subjects with antibody titers ≥1:40 before and after vaccination between children who received TIV and those who received placebo, and Wilcoxon signed-rank tests were used to compare ratios of pre- to postvaccination titers. Statistically significant differences in paired pre- versus post-vaccine responses were determined by determined by Wilcoxon matched-pairs signed rank test. For comparisons between vaccine groups and infection outcomes, a Kruskal–Wallis test with Dunns multiple comparisons was used. Outliers within longitudinal antibody analysis were excluded by the ROUT method and $Q = 1\%$ with the number of excluded outliers shown in figure legends. Longitudinal data is shown as individual fold change data with group mean fold change versus day 0 year 1 pre-vaccination response, and SD is shown (dotted lines), with statistically significant differences between V0 and V1 and between timepoints by the linear mixed model with Tukey's multiple comparison test, or V1 versus the theoretical value of 1 noted below figure plots by one sample Wilcoxon test. Correlations between NK92 cellular ADCC and FcγR3a binding and between ADCC and age were analyzed using Spearman's nonparametric correlation. For heatmap analysis of MFI (Figs. 4 and 5), upon log transformation, the data were further normalized by mean centering and variance scaling each feature using the z-score function in Matlab. Log-transformed and z-scored data were used in PCA, cluster analysis, and logistic regression models. For longitudinal fold change analysis (Fig. 2), day 30 or

subsequent year samples were divided by day 0 as the baseline, and fold changes are shown as a heatmap in Supplementary Fig. 1.

Statistical significance as shown by: $*p < 0.05$, $**p < 0.01$, $***p < 0.001$, $****p < 0.0001$, ns = not significant. For volcano plot analysis, the mean difference between vaccine groups or timepoints was compared, including correction for multiple comparisons false discovery rate (FDR), and a $q$-value greater than $-\log_{10}1.3$ was considered significant, as indicated by a dotted line for significant results by Mann-Whitney test.

## Principal component analysis (PCA) and multiple factor analysis (MFA)

PCA analysis is based on log-transformed ($y = \log(x + 1)$) and scaled input data (antibody response). The values of different antibody responses were scaled with unit variance (using pcaMethods (version 1.84.0)[43] prior to PCA analysis. The PCA results were extracted and visualized using factoextra (version 1.0.7)[44] and a self-developed R script.

MFA is a multivariate statistical technique that extends PCA and multiple correspondence analysis to simultaneously handle mixed data types. In the present study, MFA was employed to analyze the dataset containing both numerical (antibody responses and age) and categorical variables (Gender, pre-infection HAI response, and post-vaccination GMT response). Similar to the PCA analysis, the input antibody responses are log-transformed ($y = \log(x + 1)$) and scaled. The analysis was carried out using R software, specifically utilizing the FactoMineR and factoextra packages via a self-developed R script.

Similar to PCA, with the primary objective of reducing dimensionality while preserving the maximal possible variation within the data, MFA identified the optimal number of retained dimensions. Subsequently, the results were plotted as scatterplots, displaying the relationships between individual observations and the newly formed factor space.

## HA cross-reactivity

We measured the breadth of antibody responses against different HAs by calculating the HA cross-reactivity ($\pi$) in this study. The concept of HA cross-reactivity ($\pi$) is borrowed from nucleotide diversity ($\pi$), which provides an unbiased estimate of diversity among groups[45]. Specifically, we used the V0S1 pre-infection samples as the control group. The frequencies of positive HA responses (determined by whether the average responses are at least 20% higher than the average response of the V0S1 pre-infection samples) were summarized for each HA/group, and all the negative responses were characterized in a negative group. The difference values here are shown as average difference values in the $x$-axis of the Fig. 5e.

Then for every group where $n_i$ samples of RBD/negative responses $i$ are observed, RBD cross-reactivity ($\pi$) can be calculated based on pairwise difference between antigens (RBD/negative groups) as

$$\pi = \frac{\sum_{i \neq j} n_i n_j}{\frac{1}{2}N(N-1)} \tag{1}$$

where $N$ is the total number of all responses. We also calculated the classic Shannon entropy for comparison, and the results are comparable, detailed implementation of the diversity measurement can be found in codes below.

## Feature selection with Elastic Net and prediction using PLSDA

Similar to Selva et al.[46], Elastic Net was used to select the minimal set of features needed to predict infection outcomes (Supplementary Fig. 5d). The input data is 26 shared variables of 24 antibody responses and 2 types of HAI titer at pre-infection timepoints. The

24 antibody responses included four proteins (i.e., sH1-2007, pH1, sH3-2007, and H1-stem) and six antibody features (i.e., IgG, IgG1, IgG3, IgA1, FcγR3a, and FcγR2a) (Supplementary Table 2), resulting in the comparison of 24 variables. Parameters used for the model are 2000 iterations, alpha = 0.5, 10-fold cross-validation, and detailed codes are available.

**Correlation network analysis with Feature selection by Elastic Net**

All antibody features (Supplementary Table 1) at timepoint 2 of samples from sub-study (iii) or sub-study (iv) (Table 1) were assessed, and the top 11 selected variables (selected by Elastic Net, frequency >80% in 2000 iterations) to visualize their correlation with other variables as networks have done. As there are >100 different antibody variables in the data, only variables with Pearson correlation coefficient >0.8 with at least one selected variable were shown. The selected variables were highlighted in red (Fig. 5d).

**Logistic regression**

Logistic regression models for the prediction of infection were built using sub-study (iii) data (n = 115 samples) as training sets and sub-study (iv) data (n = 36 samples) as validation sets. The input variables of the predictive model are the 10 variables at post-vaccination pre-infection timepoints, the output value is whether they become infected (S1 or S0 group, infection as 1 and non-infection as 0). We choose the 10 input variables by choosing the top 10 shared variables between sub-study (iii) data and sub-study (iv) data, in terms of their ranking in the Elastic Net with PLSDA feature selection results. Specifically, each ranked list contains 24 variables (above-mentioned in the Feature selection with Elastic Net and prediction using the PLSDA section). In each round (done sequentially from rank 1 to rank 24), we compared the same-ranked variables from both lists. If they match, the variable is selected. If not, the variables are added to a waiting list. In subsequent rounds, new variables are compared to each other and those on the waiting list. Once a match is found, the variable is selected. This process continues until 10 variables are selected. Details and modeling R scripts are available.

The formula to calculate accuracy (2) is

$$\text{Accuracy} = (\text{True Positives} + \text{True Negatives})/(\text{True Positives} + \text{True Negatives} + \text{False Positives} + \text{False Negatives}) \quad (2)$$

The formula to calculate sensitivity (3) is

$$\text{Sensitivity} = \text{True Positives}/(\text{True Positives} + \text{False Negatives}) \quad (3)$$

**Reporting summary**

Further information on research design is available in the Nature Portfolio Reporting Summary linked to this article.

## Data availability

Datasets generated and/or analyzed during the current study are included in the paper or are appended as supplementary data. Source data are provided with this paper.

## Code availability

Analysis scripts have been deposited at Zenodo under accession code https://zenodo.org/records/10583684.

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

## Acknowledgements

This project was supported by Health and Medical Research Fund (17161162, S.A.V.), University of Hong Kong seed grant (201810159004, S.A.V.), Doherty Sino-Australia Partnership Travel-Research Seed Grant 2023/24 (SAV), Theme Based Research Scheme (T11-712/19-N, S.A.V. and B.J.C.) by the General Research Fund, NHMRC Investigator Fellowship (S.A.V. and A.W.C.), C2i, InnoHK, an initiative of the Innovation and Technology Commission, Government of Hong Kong SAR (M.P.) and the National Institute of Allergy and Infectious Diseases (grant no. R01 AI170116). The initial clinical trials were supported by the Research Fund for the Control of Infectious Diseases of the Health, Welfare and Food Bureau of the Hong Kong SAR Government (grant CHP-CE-03) and the Area of Excellence Scheme of the Hong Kong University Grants Committee (grant AoE/M-12/06). The funding bodies had no role in study design, data collection and analysis, preparation of the manuscript, or the decision to publish.We gratefully acknowledge the support from Vicky Fang, Mahen Perera and Samuel Cheng (The University of Hong Kong) for sample collation, and Rory De Vries and Guus Rimmelzwaan for initial discussions of the study.

## Author contributions

S.A.V., B.J.C., and N.H.L. conceptualized the study, J.Z.J., C.A.C., M.M., A.W.C., and M.P. performed experimental assays. H.G., L.L.M.P., and T.T. performed specialized data analysis. N.B. and R.V. provided specialized reagents. G.L., N.H.L., and B.J.C. conducted the clinical trial for sample acquisition. S.A.V. and B.J.C. provided funding acquisition. The project was supervised by S.A.V. The original draft was written by S.A.V., J.Z.J., C.A.C., H.G., and reviewed and edited by S.A.V., J.Z.J., C.A.C., N.H.L., B.J.C., H.G., and L.L.M.P.

## Competing interests

B.J.C. has consulted for AstraZeneca, Fosun Pharma, GSK, Haleon, Moderna, Roche, and Sanofi Pasteur. The remaining authors declare no competing interests.
