## [Peer Review File · Nature Communications]

Influenza antibody breadth and effector functions are immune correlates from acquisition of pandemic infection of childrenReviewer #1 (Remarks to the Author):

Jia et al use an existing biobank of specimens collected longitudinally from children in relationship to influenza vaccine in the context of pandemic H1N1. Overall this effort is an important investigation as our understanding of immune responses in children is limited compared to adults. Additionally, obtaining longitudinal samples from children is difficult to do making this an interesting dataset. The authors should also be commended for making the good use of limited sample volumes particularly given how precious these type of samples are in the pediatric age group.

There are some issues with way in which the manuscript is currently comprised. One major issue is the structure of the results sections. There isn't always clear delineation of where one section ends and another begins. Overall re-organization of the data would greatly benefit the readers ability to dissect the results. Some of the most interesting data comes from understanding dynamics in the Ab response following vaccination vs natural infection but isn't the focus of the results. The groups aren't well matched enough and too small to draw the type of conclusions the authors were hoping to draw which becomes apparent in their modeling at the end of the manuscript. Specific comments regarding the manuscript include the following;

1) Description of cohorts/biobank. I found it difficult to delineate the number of children in the cohort and the description makes the cohorts much more confusing than necessary. The entire paragraph from line 101-118 doesn't belong in the introduction. IT would serve the manuscript to rewrite this to make it more clear for readers and include as the first section of the results.

2) Cohorts - the vaccinated and placebo groups appear to be immunologically distinct at baseline in several of the assays making some of the claims difficult to interpret and possibly overstated. A more detailed analysis/discussions of the differences of the cohorts would improve transparency and could be included in the first section of results, similar to the suggestion above.

3) Fig 1a - this graphic doesn't fit well with the rest of the figure and likely belongs as a supplement. The inclusion of the aims feels more like a grant proposal than a figure for a manuscript. It would be good if the authors could find a better way to depict this.

4) Figure 1 and 2. Are related data but appear in different figures.

5) Figure 2. It appears that the baseline for the placebo group is higher for all but NP assay. The issue with this is that the authors use a log fold change comparison to draw conclusions regarding this assay. The appropriate comparison is already included in (A) where baseline is compared to follow-up and no change is seen in the placebo group. Did the authors attempt the same binding assay depicted in (C) for pN1?

6) Figure 3. The description of this cohort should be upfront so the readers know the sample size is small. Additionally, these results should be qualified further that its possible infection took place and wasn't reported given the inclusion based on reporting or a >4-fold raise in HAI response. To make readers more confident in this data it would be better to see each individual donor plotted longitudinally (possible in supplement). The vaccination group in particular shows some concerning dynamics as very low MFI in 2010/2011 are followed by higher levels in 2012/2013.

7) Not clear why authors chose to include 4f from only the V1S0 group.

8) Figure 4 - the age groups which are selected are arbitrary. Have the authors performed a simple linear regression of this data to see if there is a direct correlation with age?

9) Figure 5 - This is another section where the 2 groups appear quite different at baseline. An analysis determining what made these groups so distinct would benefit the manuscript.

10) Figure 5 - F-J. Its unclear anything can be called protection here. Protection would either have to be exposed but didn't develop infection (which can't really be known in humans) or infected and not severe disease (not part of authors definition).

11) Figure 6 - legend defines unvaccinated infected (V0S1, n=7 post vaccinated, n=17 post infection)

12) Figure 6 - the way the data is represented makes it very difficult to interpret. The most interesting data in 6E is intriguing but lost in the current construction of the data.

Reviewer #2 (Remarks to the Author):

The study by Jia et al. aims to address the established need for broader correlates of protection than HAI for influenza infection using a serum antibody NK cell degranulation assay, Fc receptor binding, and antibody binding breadth. A strength of the study is the unique serological cohort of children with longitudinal assessment of infection, enabling the correlation of immune parameters with susceptibility to infection. However, sample sizes are too small given the variability in the data to make strong conclusions, limiting the impact of this study. Concerns about the sensitivity and robustness of the multiplex bead binding and NK cell CD107a assays used to generate the vast majority of the data are the primary scientific concern. Highlighting the questions being addressed and when presenting the results would greatly help the reader follow the large amount of data in this manuscript.

The primary scientific concern relates to the multiplex bead binding assay. No raw data or description of how this assay was validated are presented. What does the fold-change in MFI represent in this assay? Is the assay linear across its dynamic range? How were protein stability and epitope availability validated after conjugating each antigen to the beads? How long were antigens stable on the beads during storage? Were all assays run from the same round of conjugation?

Similarly, there are concerns about the specificity, linearity, and sensitivity of the CD107a assay, though the plots showing raw data in Fig. 1b help the reader understand the assay. The authors demonstrate correlation between FcγR3a binding and CD107a signal in Fig. 1. However, a higher percentage of CD107a+ NK cells are observed for pH1 than sH1 in all groups (Fig. 2a), despite lower FcγR3a binding for pH1 than sH1 (Fig. 5d). Do the different antigens have different levels of background in this assay? Could the hemagglutinins directly interact with glycans on the NK cell line (perhaps to a different extent between strains)?

Minor points:

Fig. 1c, d: legend says linear regression, but methods say Spearman (line 480).

Fig. 2a: The lack of increased signal in this assay against vaccine antigens post vaccination is surprising. The only antigens for which increased signal is observed after vaccination are not present in the vaccine; this should be discussed in the text.

Fig. 2c: The authors explicitly mention the multiplex nature of the binding assay, but only show one protein. The CD107a assay would predict increased FcγR3a binding to pN1, even more so than to pH1 for which data are shown.

Fig. 2b-c: Was a t-test or Mann-Whitney used for comparisons between groups? Error bars hard to see in panels a and c. Would be easier to read and more consistent if they were black like in panel b.

Fig. 3: What is to be made of a statistically significant increase in H3 IgG binding in 2012 in panel c? Could this be due to subclinical (and non-seroconverting) infection? Should the statistical tests across time points be assessed using a linear mixed model? The Mann-Whitney, one-sample t, and Wilcoxon tests all assume samples are independent, and longitudinally collected samples are not. Were the tests corrected for multiple comparisons?

Fig. 4: It would help the reader to introduce these data as looking prospectively for correlates prior to infection among cohorts that would seroconvert in the next year. Why are HAI data not plotted? If the goal is to identify a superior correlate of protection to HAI, why not compare to it specifically? Also,

HAI methods are missing.

Line 184: "age related increase in pN1 ADCC responses..." is this difference statistically significant between ages?

Fig. 5e: What point is being made in this panel? Correlation between CD107a and FcγR3a binding has been shown in Fig. 1.

Fig. 6: How long post-infection (where known)?

Extended Data 4 legend is incomplete

Line 254: Times post infection for the 'd3' time point are not given (and in most cases likely not known). Since antibody titers wane rapidly from their peak, comparing the magnitude and diversity of antibody responses at unknown and variable times post infection to a peak time point post vaccination ('d2') is likely not very informative.

Is it possible the generally more robust serological profile of the V0S0 group is because it is partially or wholly comprised of non-seroconverting "infection failures" referred to on line 329 whose antibody breadth and magnitude are being boosted by subclinical infection?

Line 342: "...these cross-reactive responses are less focussed and not trained by seasonal virus exposure..." If not by exposure to seasonal viral strains, how do these cross-reactive responses arise?

Line 52: "...vaccinated children that became infected (V1S1)...made short term IgG3 with a H3-HA focus post vaccination" is contradicted by line 202 and Fig. 5c, f, and g, in which the V1S0 group has more sH3 IgG3.

P-values are missing in some locations in the text (e.g. line 170)

Reviewer #3 (Remarks to the Author):

The paper describes evaluating serological correlates of protection against pandemic H1N1 2009 illness in children in Hong Kong in archival serum samples. There are only limited studies of correlates of protection in children against influenza and this study provides important results for understanding the complexity of correlates of protection. The use of archival, serum samples collected during placebo RCTs with long-term follow-up for influenza like illness, provides the optimal study design for these evaluations. The work has focused on binding antibodies to seasonal, pandemic and avian, influenza proteins, as well as evaluating more broadly reactive ADCC responses. The cohorts are in principle, large enough to analyse for broad reactivity and the study has three specific aims. The choice of children and methodology are appropriate to meet the study aims. The data in this paper are thorough and the results are interesting. However unfortunately this paper requires a major revision to improve the readability, understanding and interpretation of the results. In its current form, it is very difficult to read, as many sentences are trying to say too much, and the key messages are missed by the reader. If the paper is read thoroughly by the authors and rewritten so that a number of sentences are divided to clearly explain the results then the paper will provide important findings for the field.

Major considerations.

- 1) the terminology used to describe the groups is difficult and it will be better if the groups are properly described in the text and particularly in the figure legends.
- 2) it is unclear how many unique children's samples are involved in this study. I think it would be helpful to include the number of children (n) in figure one under the aims and in the legends of each figure.
- 3) different sources of antigens were purchased, and it is important to include descriptions of whether the proteins are in their native form as HA trimers and NA tetramers and the aa regions of the proteins in the assays should be included in the table.
- 4) the authors often do not properly explain which children's groups are being compared, and this needs to be addressed throughout the results.
- 5) What was the likely exposure to pandemic H1N1 in the children? were there school closure or other factors which may have influenced if children were exposed which can help to define correlates of protection

6) unfortunately in its current form the paper is too difficult to read and understand due to the complexities of the sentences.

7) clear thought should be given to abbreviations and they should be consistent throughout the paper e.g. Mainly pH1N1 is used but sometimes pdm is used, pdm is the commonly used abbreviation and would make a clear distinction from seasonal H1N1

Minor consideration

Title: are they a single correlate of protection or correlates of protection?

Abstract: structuring the abstract into introduction methods, results and conclusions may improve the readability. Please include how many children are included in the study, and that long-term follow-up for five years. Please define abbreviations upon first use. I would avoid using the VO, V1, SO, S1 abbreviations in the abstract (and the whole paper) as it unnecessary.

Line 33 missing a word resulted in partial vaccine effectiveness.

Lines 34-37 perhaps split into two sentences. Also residual protection needs to be explained

Line 37 several places throughout the paper term immune serum is used, serum is sufficient

Line 40 RT PCR is also used to define infection, not only sero conversion

Line 43 isotypes should come before subclasses, IgA1 is not an isotype.

Line 44 the study only had day 30 and one year follow-up samples, so it would be better to say seasonal vaccination increased H1N1 specific antibodies which declined at 1 year

Line 48 which other groups?

line 50 in their antibody responses instead of to

Line 52 which other groups? Also short term you did not have samples between day 30 and 1 year

Line 55 HAI remove antibody as the definition is incorrect

Line 57 what are FcR binding features?

Line 60 include H1N1

Line 65 this is incorrect elderly and people with comorbidities experience the highest hospitalisations.

Line 66 can you rephrase this e.g. are not designed to protect

Line 69 divide into two sentences,

Line 75 against pandemic infection, this sentence does not make sense and again should probably be split into two sentences.

Line 77 perhaps change to limiting infection (instead of acquisition) and severity of disease?

Line 84 subtypes and strains (in order of diversity)

Line 85 perhaps include influenza A H1N1, H5N1 and H7N9.

Line 85 Lethal- they are commonly referred to as highly pathogenic

Should the structure of stem and division into two groups for Influenza A viruses be explained in the introduction?

pH1N1 or pdm (see 7 above), pdm would make a clear distinction from seasonal H1N1

Line 91 abbreviate Matrix protein which is used later.

Line 91 Lending should this be allowing. Is a word missing? To provide greater subtype...

Line 96 is the half-life of human IgG3 only 1 week?

Lines 93-98 divide into at least 2 clear understandable sentences. What does potency with time since exposure mean?

Line 101 remove immune

Line 103 to assess influenza specific antibody breath.

Line 103 -108 divide into short sentences that explain the study background

Line 109 change to children were monitored for influenza H1N1 infection. Define RT-PCR

Line 110 change to serologically

Line 112 RCT is now described as a cohort study

Line 114-5 include avian H5 and H7 viruses

Line 116 describe 3 aims in introduction

Line 117 delete to be made for

lines 119-122 separate in short clear sentences explaining the aims. Here pandemic H1 N1 infection is

described as pdmH1N1, not pH1N1.

Line 125 remove one of cell in title. Again this is heading is trying to say too much in one sentence. Focus on the important findings.

Line 127 a brief introduction to the samples and the study aims would be a improve the sart of the results section.

Lines 128-132 again, please split the sentence into the methods and then the findings. And the correlation show binding antibodies versus degranulation.

Figure 1a I think this figure needs more thought to clearly explain the study set up

- please include the number of children in each aim and that the follow up samples are collected yearly for 5 years.

- Aim 1.1 does not make sense. Seasonal vaccination for induction of pandemic ADCC. Aim 1.2 longitudinal stability of what?

- Could aims 1.1 and 1.2 be combined into one aim?

e.g. Impact of seasonal vaccination on Pandemic ADCC and durability

- Aim 3 should this be HA diversity of antibody responses

- There are a lot of undefined abbreviations in the figure legend.

- Can you clearly state blood samples were collected baseline (day 0), day 30 (post vaccination) at 1 year follow up in 2010 and for a subgroup (n=x) yearly between 2010 and 2014.

- Think about the use of brackets in the figure legend

- Delete proteins of interest

Line 134 divide sentence into two, e.g. although postvaccination a significant increase in fold change for pN1 responses was observed compared to placebo only. (you can say, placebo only and delete controls)

Figure 2c is the original data presented in the paper or just fold change. It appears there only six individuals who have higher responses. Have you conducted a sensitivity analysis to show if these individuals are unduly influencing the results?

Lines 137-140 can you simplify this Having confirmed the correlation between multiple binding to NK cellular functional degranulation, multiplex assays were used to evaluate multiple responses which is ideal for paediatric samples as it only requires low serum volumes.

Line 142 is durability a better word than longitudinal stability? Should this read after seasonal vaccination instead of by vaccination? Perhaps start with this is a small subgroup of 14 subjects.

Table 1 is the use of mean age suitable for children, often median age is used in paediatrics

Line 144-145 move to discussion as not relevant for children

Lines 146-151 divide into shorter sentences

Line 149 Matrix protein has not been previously defined

Lines 151-156 this is incomprehensible sentence, please split into shorter sentences

Lines 162-166 perhaps shorter sentences and define the four groups at the start

Figure 2

- It would be helpful to have placebo and vaccination as subheadings in the figure legend

- be consistent in the terminology used to describe the NK cell ADCC

- c I would include placebo and vaccination on x axis

- Figure 2c where is the original data. It appears there only six vaccinated individuals who have higher responses. Have you conducted a sensitivity analysis to show if these individuals are unduly influencing the results?

- There are a lot of undefined abbreviations in the figure legend.

- Think about the use of brackets in the figure legend

Figure 3a can you change to show yearly samples between 2010- 14

Figure 3 The ordering of graphs is strange should NP (b) be beside M1 and stem together with H1

Table 1 the symptom score should be defined in the legend and do you need to separate this into two lines of the table? If so clearly define each as refereeing to either RT-PCR or serologically confirmed infection.

Can children be in several groups or are they only in one group?

Table 2 and Table 3 do these need to be a main tables or can abbreviations be included into the figure

legends?

Lines 174-177 the rationale for using HAI titre of 40 as detectable titres needs to be included as detectable titres are often 10 or 20 (depending upon the stating dilution), whilst 40 is considered protective in adults. Some of the text from the paragraph on HI titres for H3N2 in the discussion could be used. Normally HAI are presented as GMT for each group.

Line 183 yo is missing after 9-11

Line 188 can you write this more clearly e.g. The Multiplex assay was expanded to measure both binding (isotype/subclass) and functional binding antibodies against pandemic, seasonal and cross reactive proteins.

Line 192 change to simplify however, no differences....

Line 194 is this correlation with the same children?

Line 196 change per to for

Line 197 which is befitting perhaps change to which is as expected.

Line 200 can you simply say Multiplex comparisons of post vaccination

Line 204 the sentence refers to studies but only reference 14 is provided (also is this a discussion point)

Line 204 to 205 simplify to Whilst comparison of infected and uninfected vaccinated children

Line 207 simply put the p value next to the significant values.

Line 220 how many samples were paired? Include n

Line 227-229 rewrite the sentence as it is difficult to understand

Line 237 why is d2 timepoint introduced here and can you simply give the days post vaccination. This applies to all subsequent descriptions. I think you are trying to say H3 antibody responses correlates with Igs and FcR, but this was not observed for other parameters.

Line 254 d1 versus d2 has not previously been defined, perhaps use pre-to post

Discussion: The discussion could do with being better structured lines, focusing first on the main findings, moving the limitations to the second from last paragraph and ending with conclusions.

Lines 288 to 304 are more background. Perhaps the section on HAI antibodies should be moved to the introduction

Lines 318-319 move specific antibodies to after the FC region, change to due to limited archival sample volume

Lines 331-335 could be maybe moved to the start of the discussion at the end of the first paragraph

Line 349 delete were then

In children protective IFN- γ T cells have been found after LAIV and should be mentioned in the discussion (PMID: 18448618)

Data availability, the data should be supplied as deidentified for each figure/ table. Or at the very least the contact for the data should be included

Methods

Line 386 was vaccination conducted in 2008 or 2009, as the vaccine used was the 2009-10 TIV

line 392 delete one of based

line 397 did both parents provide informed consent or only 1?

Line, 433 incubate should this be in the past tense? Do you mean sera?

The HAI methodology is missing. Why have the authors chosen to use a titre of 1:40 as positivity? Is this because it has been associated with protection in adults?

Line 493 Missing upon log transformed

Line 498 change to we measured

Reviewer #4 (Remarks to the Author):

This manuscript reports on an analysis of immunological correlates of protection from influenza infection, using samples from a historical placebo-controlled trial of seasonal influenza vaccination. Seasonal influenza vaccination was found to increase pH1N1 specific antibodies and FcR effector functions, which declined over 1 year post-vaccination. Antibodies and FcR effector functions post-

vaccination were not clear correlates of influenza infection among the vaccinated, although pH1 and pN1 ADCC antibodies were higher in magnitude among unvaccinated uninfected vs. infected individuals.

The statistical methodology is difficult to follow, and in some instances overlooks issues of confounding and multiple comparisons and does not address the stated objectives of the study. In particular,

- basic descriptive statistics are missing, eg how many cases and controls were tested at each time point, what is the overall level of efficacy of the seasonal vaccine, how many cases and controls in training and test sets, how many subjects for addressing each aim
- Objectives are not laid out clearly enough (Figure 1a)
- Lines 160 and following, where the immune correlates analysis is reported, unclear why a Kruskal-Wallis test is used as the contrasts are between two groups
- Figure 3, unclear if there is adjustment for the multiple comparisons that are done, and why there are tests at each time point rather than tests for trends over time which would be more powerful
- Figures 4 and 5: some of the contrasts are uninterpretable. Eg why contrast vaccinated infected individuals (V1S1) with unvaccinated uninfected individuals (V0S0)? The objective here is to understand predictors of infection, which involves the contrast between V1S0 and V1S1, and between V0S0 and V0S1.
- It is not clear if any covariate adjustment was done when contrasting infected vs. uninfected individuals, beyond the matching for age. A rigorous analysis would include adjustment for potential predictors of immune response and risk of infection, to control for confounding.
- Description of the predictive model evaluation on lines 269-73 is especially difficult to parse; "accuracy" and "sensitivity" metrics are not defined
- I do not know how to interpret the analyses shown in Figure 6

The small size of the study limits the strength of the conclusions. The longitudinal stability of the antibody responses is especially imprecise given just 14 participants.

More minor comments:

The design of the study needs to be more clearly described prior to presenting the results; the description on lines 112-116 is inadequate. Eg stating that a matched case-control study was used, and how many cases and controls were selected and tested at each time point.

"Baseline" term is confusing as this refers to Day 30, not Day 0

"Experiment was repeated twice" is unclear: were there duplicate samples?

Reviewer #1 (Remarks to the Author)

Jia et al use an existing biobank of specimens collected longitudinally from children in relationship to influenza vaccine in the context of pandemic H1N1. Overall, this effort is an important investigation as our understanding of immune responses in children is limited compared to adults. Additionally, obtaining longitudinal samples from children is difficult to do making this an interesting dataset. The authors should also be commended for making the good use of limited sample volumes particularly given how precious these type of samples are in the pediatric age group.

There are some issues with way in which the manuscript is currently comprised. One major issue is the structure of the results sections. There isn't always clear delineation of where one section ends and another begins. Overall re-organization of the data would greatly benefit the readers ability to dissect the results. Some of the most interesting data comes from understanding dynamics in the Ab response following vaccination vs natural infection but isn't the focus of the results. The groups aren't well matched enough and too small to draw the type of conclusions the authors were hoping to draw which becomes apparent in their modelling at the end of the manuscript.

Specific comments regarding the manuscript include the following;

R1.1) Description of cohorts/biobank. I found it difficult to delineate the number of children in the cohort and the description makes the cohorts much more confusing than necessary. The entire paragraph from line 101-118 doesn't belong in the introduction. It would serve the manuscript to rewrite this to make it more clear for readers and include as the first section of the results.

A1.1. We appreciate the reviewers comment, for clarity we have included group sizes in each figure at the timepoint sampling descriptor, in the legend text and at Table 1.

Section lines 101 to 118 has been moved from the introduction to the results section where it more appropriately belongs to introduce and describe the study cohort.

Additional sample information for the samples used is now included as supplementary data for each figure panel including individuals age, gender, HAI status, vaccination and infection status for each group, for which the group averages are summarised in Table 1.

R1.2) Cohorts - the vaccinated and placebo groups appear to be immunologically distinct at baseline in several of the assays making some of the claims difficult to interpret and possibly overstated. A more detailed analysis/discussions of the differences of the cohorts would improve transparency and could be included in the first section of results, similar to the suggestion above.

A1.2. We agree with the reviewer, and have provided many descriptors of the cohort, in terms of age, gender, HAI and timepoints in Table 1. HAI data has also been included with each sub-study of samples also. However, baseline differences, i.e. post vaccination but pre infection, may be different between groups as these children also had subsequent different outcomes of seroconversion and therefore risk of infection. Children were initially randomised for vaccination, but post vaccination and baseline pre-infection differences may be apparent which determine infection outcomes, and are the basis of this study.

A Multi Factor Analysis based on Age, Gender, Vaccination, HAI status and Antibody measures was also conducted (see R1.9) and found that age and gender contributed less to group differences than antibody parameters. We have included the limits of confounders- such as social contact now in the discussion.

We have updated the first results section to include a description of Table 1 for their age gender and HAI results as a first figure panel for cohort of samples.

In the first results section, page 4, we have added a more detailed description of the cohort: “In this current study, participants were selected (Table 1) from this initial larger RCT study from various timepoints on the basis of seasonal vaccination (V1) or placebo (V0), and pH1N1 infection inferred by sero-conversion (S1) or no seroconversion (S0) against pH1N1 by HAI (>4 fold rise) within the first year of the study. There were 4 possible types of samples used, V1S1 (vaccinated infected), V1S0 (vaccinated uninfected), V0S1 (unvaccinated infected) and V0S0 (unvaccinated uninfected). Children were selected for 4 different sub-studies for antibody studies, (i) to determine vaccine induced ADCC changes (Fig. 1), (ii) longitudinal stability of vaccine induced antibody FcR binding and isotype changes (Fig. 2), (iii) differences baseline FcR binding and isotype with acquisition of pH1N1 infection (Fig. 3 and 4), and (iv) the diversification of HA-specific antibody responses with vaccination and infection (Fig. 5). The group size, age and gender were comparable between sub-studies, as uninfected controls were selected to match infected children. The majority of vaccinated children seroconverted (by HAI for >4 fold rise) to sH1 (55-90% of each sub study V1 group) but not pH1 (0-17% of each sub study V1 group), whilst fewer placebo children seroconverted to sH1 (0-12% of each sub study V0 group) and pH1 (6-20% of each sub study V0 group) (Table 1), consistent with previous reports of the larger cohort¹. Whilst the frequency of participants that had high baseline (>40 HAI titer) (after vaccination, pre infection) pH1 responses trended to be lower in V1S1 and V0S1 groups, but were comparable for sH1 across the 4 groups (Table 1). Thus, differences in baseline HAI exist in the cohort that are related to infection outcomes and determining these in finer detail are the focus of this study.”

R1.3) Fig 1a - this graphic doesn't fit well with the rest of the figure and likely belongs as a supplement. The inclusion of the aims feels more like a grant proposal than a figure for a manuscript. It would be good if the authors could find a better way to depict this.

A1.3. We appreciate the reviewer's comment and have adjusted Figure 1a for clarity to remove labels of sample selection, which is covered instead within each subsequent figure panel and in Table 1 for the sub-study samples. We have relabelled 'Aim' to sub-study and figure numbers in Table 1 where the participant details are described.

R1.4) Figure 1 and 2. Are related data but appear in different figures.

A1.4. Figure 1 and 2 are now combined and redundant data of Figure 2 removed for clarity.

R1.5) Figure 2. It appears that the baseline for the placebo group is higher for all but NP assay. The issue with this is that the authors use a log fold change comparison to draw conclusions regarding this assay. The appropriate comparison is already included in (A) where baseline is compared to follow-up and no change is seen in the placebo group. Did the authors attempt the same binding assay depicted in (C) for pN1?

A1.5. We appreciate the reviewer's comment, and to clarify there are no significant baseline differences in Figure 2 (previous version, now figure Figure 1c) between placebo and vaccinated groups, this is now stated in the results:

Page 5, Line 254: “Seasonal vaccination (V1) increased NK cell ADCC responses for pN1 ($p=0.008$) and pH1 ($p=0.02$) (Fig. 1c) at day 30 from day 0 responses, whilst there was no difference in functional ADCC responses between V1 and V0 groups at day 0.”

We have removed unnecessary panels from figure 2, and combined pre/post vaccination NK92 cells responses to Figure 1. The original panel figure 2c has been removed as it was only conducted for pH1 protein, not pN1, as part of the assay transition from ELISA to Luminex bead approach for FcR binding.

R1.6) Figure 3. The description of this cohort should be upfront so the readers know the sample size is small. Additionally, these results should be qualified further that its possible infection took place and wasn't reported given the inclusion based on reporting or a >4-fold raise in HAI response. To make readers more confident in this data it would be better to see each individual donor plotted longitudinally (possible in

supplement). The vaccination group in particular shows some concerning dynamics as very low MFI in 2010/2011 are followed by higher levels in 2012/2013.

A1.6. We appreciate the reviewers comments, and agree that group size is an issue in paediatric studies and our stringent longitudinal criteria of no further vaccination or infection during the study period limited our sample size. Group size from the legend is also now given in the figure 3a panel (n=6 unvaccinated, n=8 vaccinated), and the description of these results ends with the statement:

“Children were also selected based on the criteria of no further vaccination or infection (by seroconversion) during the subsequent 5 years (Fig. 2), resulting in a limited sample size of V1 (n=8) and V0 (n=6) due to stringent criteria, which precludes strong conclusions in antibody durability.”

We have also added to the results section (page 5, line 267) when introducing the sub study (ii) longitudinal cohort:

“We are unable to exclude the possibility if infection occurred between monitoring periods that was asymptomatic and resulted in less than a 4-fold HAI response, however this remains unlikely based on baseline HAI titers and GMT rises (Table 1) and no changes in NP-specific antibody responses.”

We have updated Figure 2 (previously Figure 3) to show individual responses, however if we included plots for each donors for each protein and detector this would lead to over 140 panels, but as the groups leads to statistically significant differences we have shown this instead. We have updated the data to exclude outliers, and individual data points are shown. Figure 3 legend updated to include:

“Outliers were excluded using ROUT identification of outliers at Q=1% leading to exclusion of n=1 in V0 NP IgG, n=2 in V0 H1stem IgG, n=1 in V1 M1 IgG, n=1 V0 and n=1 V1 pH1 IgG1, and n=6 V0 and n=1 V1 pH1 IgG3.”

The exclusion of outliers has not impacted statistical differences between 2011 and 2012. The most meaningful differences are within the first year, and are the focus of the description of results.

R1.7) Not clear why authors chose to include 4f from only the V1S0 group.

A1.7. For clarity, we have removed to correlation between proteins for this group which aimed to demonstrate when responses were present that they are cross reactive.

R1.8) Figure 4 - the age groups which are selected are arbitrary. Have the authors performed a simple linear regression of this data to see if there is a direct correlation with age?

A1.8. We apologise for this rudimentary stratification and have changed the analysis to simple linear regression graphs which are updated below, which still shows the significant negative correlation of the V1S1 group for pN1 responses.

From :

To:

R1.9) Figure 5 - This is another section where the 2 groups appear quite different at baseline. An analysis determining what made these groups so distinct would benefit the manuscript.

A1.9. (Now Figure 4) The samples included are post vaccination in the vaccinated group, hence their baseline differences are higher than the placebo group, and they were prospectively assigned to S1 and S0 groups based on acquisition of infection. A comparison of the groups from Figure 3/4 and Figure 5 by Multi Factor Analysis (MFA), showed that the V0 and V1 groups are not significantly different (points overlapping with each other) even though we included Age, Gender, HAI, GMT and Antibody into the analysis, suggesting baseline differences are not significant.

1. Contributions of variables to the first 2 dimensions of MFA results
 - a. **Figure 3/4** data

group	Dim.1	Dim.2
Age (N=1)	15.20	5.58
Gender (N=1)	0.05	7.62
HAI (N=2)	34.73	36.71
GMT (N=2)	34.03	32.81
Antibody (N=56)	15.99	17.29

- b. **Figure 5** data

group	Dim.1	Dim.2
Age (N=1)	13.18	22.36
Gender (N=1)	9.83	0.23
HAI (N=2)	27.16	19.64
GMT (N=2)	29.71	25.22
Antibody (N=133)	20.11	32.55

We see Age and Gender all contribute less compared to Antibody or HAI or GMT.

2. Baseline difference between V0 and V1 groups.
 - a. **Figure 3/4** data:

b. Figure 5 data:

Reviewer figure 1: Multi factor analysis of sample groups from Figure 3, 4 and 5.

R1.10) Figure 5 - F-J. Its unclear anything can be called protection here. Protection would either have to be exposed but didn't develop infection (which can't really be known in humans) or infected and not severe disease (not part of authors definition).

A1.10. We agree with the reviewer's definition of protection as, "protection from disease", rather than acquisition of infection, and severity by hospitalization was not apparent in these community acquired cases of pandemic influenza infection in school aged children.

Therefore, throughout the manuscript we have removed 'protection' and replaced with 'acquisition of infection'.

In the discussion (page 11) we also had stipulated:

"As our study used samples that were from community acquired pdmH1N1 infection and no reported hospitalization, our study is of immune correlates of protection from acquisition of infection, not disease severity."

R1.11) Figure 6 - legend defines unvaccinated infected (V0S1, n=7 post vaccinated, n=17 post infection)

A1.11. Corrected to n=7 pre infection, n=17 post infection.

R1.12) Figure 6 - the way the data is represented makes it very difficult to interpret. The most interesting data in 6E is intriguing but lost in the current construction of the data.

A1.12. We agree with the reviewer this is very interesting data set with many nuances between timepoints and groups, but also complicated by the many factors shown. We have presented the data in multiple ways to engage the reader with the data, as a heat map (b), PCA (c) and as a 2-D representation (e) to compress the different HA proteins tested as the one measurement of HA-cross reactivity (π to represent diversity). For clarity, we have expanded the description of the results and discussion in this area.

From:

“From our heatmap and PCA analysis, the HA protein responses were compressed to gauge the extent of diversity and magnitude of antibody responses across various features to a 2-D plot versus V0S1 pre infection samples as the comparator for average difference between groups and timepoints (Fig. 6d). Across IgG1, Fc γ R2a and Fc γ R3a, the V0S0 antibody profile was the most diverse with the highest cross-reactivity score (π), and magnitude above V0S1 pre infection responses, and was second highest following V1S0 d2 samples for IgG. Following which V1S0 post vaccination responses tracked similarly to V0S0 profiles, with the addition of increased IgA1 magnitude. Breakthrough infections, i.e., V1S1 children had the largest IgM and IgG3 responses, the magnitude and diversity of which were increased by vaccination (d1 versus d2), and reduced by infection (d3). Whilst post infection (d3) V0S1 children had increased Fc γ R2a and Fc γ R3a profiles, which may achieve similar levels to V0S0 children which may provide future breadth of protection.

To:

“From our heatmap and PCA analysis patterns were apparent in the breadth of HA-specific antibody responses by vaccination and infection between groups. Therefore, a 2-D representation was created of responses of different groups and timepoints to compress for HA cross-reactivity (measured by π) versus V0S1 pre-infection samples (day 30 post vaccination timepoint, selected as the most susceptible group to infection) (Fig. 5e). The y-axis represents the diversity of the HA response, and the x-axis represents the magnitude difference from V0S1 day 30 of other timepoints or groups responses. For the detectors of IgG1, Fc γ R2a and Fc γ R3a, the V0S0 group had the most diverse antibody response with the highest cross-reactivity score (π), and a magnitude response that was significantly above V0S1 pre infection responses. The V1S0 post vaccination samples showed similar patterns to V0S0, with the additional feature of significantly increased IgA1 magnitude. Whereas, breakthrough infections, i.e., V1S1 children had the largest IgM and IgG3 responses of the study, and the magnitude and diversity of these responses were increased by vaccination (day 0 versus day 30), and were reduced following infection (year 1). Whilst V0S1 children at post infection (1 year) acquired increased Fc γ R2a to a greater extent than Fc γ R3a responses compared to their pre infection (day 30) responses, which may provide future breadth of immunity.”

Which is also further discussed, and has been modified for clarity, at page 11:

“Considering the 2D representation of responses, the magnitude and diversity of IgG1, Fc γ R2a and Fc γ R3a responses was highest in V0S0 children than in other groups suggesting features of their antibody profile are more cross reactive and with effector functions. Antibody responses in V1S0 children post-vaccination attained some of these functions and levels, but not the same extent as V0S0 children, however they also had the highest IgA1 responses, which may contribute to their lack of pH1N1 infection. Whilst V1S1 children at post-vaccination pre-infection timepoint samples, had elevated IgG3 and IgM responses, which compete with IgG1 for Fc receptors and IgM points to a less differentiated response with lower class switching. Whilst IgG3 and IgM in V1S1 children were then decreased post-infection, there was not an accompanying increase in HA-diverse IgG1 or Fc γ R2a and Fc γ R3a binding responses but rather a reduction, suggesting that V1S1 children did not become V0S0-like in their responses post-infection and remain unable to fully maximise antibody effector functions. Thus, vaccination and prior infection cannot account for the lack of infection in V0S0 children or susceptibility of V1S1 children, and further host factors (reviewed in²⁶) such as genotype for key immune pathways (e.g. IFTIM3, CD55, IL28B) or antigenic gaps in prior immunity (no previous or recent H1N1 exposure) should be investigated²⁷. “

Reviewer #2 (Remarks to the Author)

The study by Jia et al. aims to address the established need for broader correlates of protection than HAI for influenza infection using a serum antibody NK cell degranulation assay, Fc receptor binding, and antibody binding breadth. A strength of the study is the unique serological cohort of children with longitudinal assessment of infection, enabling the correlation of immune parameters with susceptibility to infection. However, sample sizes are too small given the variability in the data to make strong conclusions, limiting the impact of this study. Concerns about the sensitivity and robustness of the multiplex bead binding and NK cell CD107a assays used to generate the vast majority of the data are the primary scientific concern. Highlighting the questions being addressed and when presenting the results would greatly help the reader follow the large amount of data in this manuscript.

R2.1) The primary scientific concern relates to the multiplex bead binding assay. No raw data or description of how this assay was validated are presented. What does the fold-change in MFI represent in this assay? Is the assay linear across its dynamic range? How were protein stability and epitope availability validated after conjugating each antigen to the beads? How long were antigens stable on the beads during storage? Were all assays run from the same round of conjugation?

A2.1. We apologise for the reviewer's confusion on the Luminex assay used to derive antibody binding responses.

- To clarify, the fold change in mean fluorescence intensity (MFI) represents the change in response pre- versus post-vaccine (such as longitudinal data in Figure 2). The MFI reflects the analyte concentration, i.e. quantity of a protein specific antibody, which is detected by fluorescent detector-PE signal which is linked to the feature of the antibody it detects, and the protein it is specific to which is coupled to a bead of a particular size/density.

From Biorad on detection of beads and signal intensity:

<https://www.bio-rad.com/en-au/applications-technologies/multiplex-immunoassays?ID=LUSM0E8UU>

“The technology enables multiplex immunoassays in which one antibody to a specific analyte is attached to a set of beads with the same color, and the second antibody to the analyte is attached to a fluorescent reporter dye label. The use of different colored beads enables the simultaneous multiplex detection of many other analytes in the same sample. A dual detection flow cytometer is used to sort out the different assays by bead colors in one channel and determine the analyte concentration by measuring the reporter dye fluorescence in another channel”.

- The multiplex assay is not linear across its range. The responses measured by this assay are not linear, which is why data is log transformed before z scoring when comparing response (as per heat map data), and between protein comparisons are not made (e.g. sH1 vs pH1), only between group (e.g. V1 vs V0) or timepoint data (e.g. pre vs post).
- Protein coupled beads are stable as they are covalently linked. Stability within 1 month is also evident when we ran samples from the same bead set at 2 machines within one month (Figure below), with no significant difference between of MFI of paired samples detected using the same coupled beads 1 month apart, indicating that this is a stable and reliable system within at least 1 month. Given that all the coupling, incubation and acquisition of samples on the FlexMap of each sub study and cohort were done in less than 1 month, we are confident to say that no degradation or decoupling of the proteins on the beads compromised the data we have presented. The assays were also all run on the same bead coupling stock, as this generates enough for 1,000 tests.

Reviewer figure 2: Stability of coupled beads fresh and 1 month later.

- The protein stability and epitope availability were measured to determine the coupling efficiency, using the signal detected by representative beads using by anti-His-PE detection (Reviewer figure 3), according to the protocol of Lopez et al., JCI Insight 2021 (<https://insight.jci.org/articles/view/150012/sd/1>). The coupling efficiency of each recombinant HA protein was measured on the multiplex assay with 10µg/ml of anti-His Tag antibody. Coupling of each variant was compared to pH1. Variants with a calculated >100% coupling efficiency to pH1 are capped to 100%, and dotted line represents 50% coupling. Experiment was repeated twice. Data represents average and standard error. Further comparison of pooled buffy positive control serum (collected 2015-2018) for beads that showed <50% coupling, showed MFI comparable to pH1, and in line with expected positivity rates (such as avian HA proteins H5 and H7 which are lower than seasonal viruses), however one protein B/Vic-2008 had low coupling efficiency and low results from pooled plasma across all detectors, results from sub-study (iv) have had B/Vic-2008 HA results now excluded (Review figure 4). No broadly reactive monoclonal antibody that covers both influenza A viruses of group 1 and 2 HA, was available to us to test for comparable MFI across the beads. However, protein to protein differences are not compared in this paper, but vaccination and infection groups are our focus. All data is log-transformed and z-scored. The 2D representation of the HA diversity response has been updated to reflect log transformed z-scored data due to protein coupling efficiency.

Reviewer figure 3: Coupling efficiency of beads detected by anti-His-PE.

Reviewer figure 4: Comparison of lower coupled beads (<50%) MFI relative to pH1 for pooled buffy serum.

The methods section has been updated to include details :

“Serum samples were run in duplicate. All assays were run on the same conjugation round within 1 month, with no degradation evident due to covalent linkages of proteins and beads. Protein conjugation efficiency was confirmed by anti-His-PE detection (ref <https://insight.jci.org/articles/view/150012/sd/1>), and the MFI confirmed by pooled buffy pack serum in the expected range. Proteins that had both lower than 40% coupling efficiency and MFI relative to pH1 across a range of detectors were excluded (data from one protein HA B/Vic-2008 was excluded). Due to differences in protein quality, conjugation efficiency, no direct comparisons are made between protein specific responses, and data is log-transformed then z-scored for heat map, PCA, and combined data analysis.”

R2.2) Similarly, there are concerns about the specificity, linearity, and sensitivity of the CD107a assay, though the plots showing raw data in Fig. 1b help the reader understand the assay. The authors demonstrate correlation between FcγR3a binding and CD107a signal in Fig. 1. However, a higher percentage of CD107a+ NK cells are observed for pH1 than sH1 in all groups (Fig. 2a), despite lower FcγR3a binding for pH1 than sH1 (Fig. 5d). Do the different antigens have different levels of background in this assay? Could the hemagglutinins directly interact with glycans on the NK cell line (perhaps to a different extent between strains)?

A2.2. We agree there is a discordance between the cellular CD107a+ NK cells and FcγR3a responses for sH1 versus pH1, but also the strength of the assay correlation is protein dependent- as shown for NP ($r=0.6028$) and pH1 ($r=0.7450$). But this is also why we do not compare between protein specific responses, instead only between vaccine/infection groups and timepoints of the same protein-specific response, furthermore this is why data is log transformed and z-scored in further analysis. For the NK92 assay, background was subtracted per participants immune serum against blocked treatment (by Fetal Bovine Serum (FBS)) from influenza protein responses, and a further internal negative control included non-specific HIV protein gp120. Without the presence of immune serum, the expression of CD107a+ is extremely low against recombinant proteins, therefore non-specific protein controls were used to ensure wash steps were adequately removing non-specific immune serum.

The reviewer raises an important future area of research in the ability of post-translational modifications such as glycans to modulate NK cell functions and FcR binding. In this case both proteins, pH1 and sH1 came from a mammalian expression system HEK293 (listed in Extended data Table 2) and thus these features may be inherent to the viral sequence, rather than the expression platform.

Minor points:

R2.3) Fig. 1c, d: legend says linear regression, but methods say Spearman (line 480).

A2.3. The data is presented by linear regression and statistical method is Spearman correlation.

The legend has been updated to read: "Lines represent trend by linear regression, and 95% confidence interval. Significant correlations measured using Spearman's correlation with r and p values shown."

R2.4) Fig. 2a: The lack of increased signal in this assay against vaccine antigens post vaccination is surprising. The only antigens for which increased signal is observed after vaccination are not present in the vaccine; this should be discussed in the text.

A2.4. Indeed, we agree with the reviewer and this forms the premise of the study. Children by traditional HAI measurements increase their sH1N1-specific antibodies but not pH1N1-specific antibodies (new Figure 1b), whilst they significantly increase ADCC function to pH1 and not sH1 by vaccination (Figure 1d).

A discussion of this is included at page 11:

"Seasonal vaccination did not bolster FcR effector functions to corresponding seasonal-specific antibody responses, which may be due to maximised FcR functions from previous encounters with related seasonal H1N1 strains. Whilst gains in pandemic-reactive FcR functions maybe attributable to vaccination briefly increasing different B cell sources. These features could be deciphered at the clonal level if cells were available."

R2.5) Fig. 2c: The authors explicitly mention the multiplex nature of the binding assay, but only show one protein. The CD107a assay would predict increased FcγR3a binding to pN1, even more so than to pH1 for which data are shown.

A2.5. Fig. 2c was done by single ELISA for FcγR3a binding, not bead based multiplex assay, when the technique was being established and not done for other proteins such as pN1. This Fig. 2bc panel has been removed for clarity.

R2.6) Fig. 2b-c: Was a t-test or Mann-Whitney used for comparisons between groups? Error bars hard to see in panels a and c. Would be easier to read and more consistent if they were black like in panel b.

A2.6. This Fig. 2bc panel has been removed as clarity to streamline data was recommended.

R2.7) Fig. 3: What is to be made of a statistically significant increase in H3 IgG binding in 2012 in panel c? Could this be due to subclinical (and non-seroconverting) infection? Should the statistical tests across time points be assessed using a linear mixed model? The Mann-Whitney, one-sample t, and Wilcoxon tests all assume samples are independent, and longitudinally collected samples are not. Were the tests corrected for multiple comparisons?

A2.7. We appreciate the reviewer's statistical expertise and have updated accordingly. The figure 2 legend has been updated for statistical tests used to exclude outliers and mixed linear models when comparing vaccinated (V1) to unvaccinated (V0) children:

"Outliers were excluded using ROUT identification at Q=1% leading to exclusion of n=1 in V0 NP IgG, n=2 in V0 H1stem IgG, n=1 in V1 M1 IgG, n=1 V0 and n=1 V1 pH1 IgG1, and n=6 V0 and n=1 V1 pH1 IgG3. Statistically significant differences shown above (by mixed linear model with Tukey's test correcting for multiple comparisons, between years* coloured for comparison group V1 blue, V0 red) and in tables below versus V0 group (by mixed linear model with Tukey's test correcting for multiple comparisons) or theoretical value of 1 (for no fold change, by one sample Wilcoxon test)."

The review raises an intriguing possibility, whilst changes in 2012/2013 could be attributable to subclinical and non-sero converting infection, they are unable to be identified in this study with no reported ILI or HAI fold rise during sampling periods. Our discussion does not focus on later timepoints fluctuations given the small sample size, and our main point is that responses wane within one year of vaccination. Further waning was evident in the unvaccinated children using the updated mixed linear model.

R2.8) Fig. 4: It would help the reader to introduce these data as looking prospectively for correlates prior to infection among cohorts that would seroconvert in the next year. Why are HAI data not plotted? If the goal is to identify a superior correlate of protection to HAI, why not compare to it specifically? Also, HAI methods are missing.

A2.8. We appreciate the reviewers suggestion, and have quoted “looking prospectively for correlates prior to infection among cohorts that would seroconvert in the next year” in the abstract, results from Figure 4 (now Figure 3) and discussion section.

Results, page 4:

“To assess prospectively for antibody correlates prior to infection, we selected children known seroconvert (S1) to pH1N1 in the next year or not (S0), based on their seasonal vaccine status. These prospective baseline samples were assessed in a NK cellular function assay for ADCC responses against pH1N1 and seasonal vaccine representative proteins (Fig. 3).”

At the reviewer’s suggestion, we have now included HAI data that was originally reported (see Cowling CID 2010 and 2012) for our study subsets. For example, now at Figure 1b:

To ensure our methods are succinct as HAI is a standardized method and previously reported, this is now referred to in the methods.

Methods, page 12:

“To assess antibody features from seasonal vaccination that may contribute to reduced or delayed acquisition of pandemic H1N1 infection, we selected archived samples (Table 1) from two trials archives (NCT00792051) previously conducted by our group in school aged children (6-15 years of age) which reported Hemagglutinin antibody inhibition (HAI) titers^{1,13}.”

R2.9) Line 184: "age related increase in pN1 ADCC responses..." is this difference statistically significant between ages?

A2.9. In line with Reviewer comment 1.8, we have removed ‘arbitrary age groups’ and linear regression of age with protein specific ADCC function, showed only V1S1 children had a significant reduction in pN1 responses with age (simple linear regression with Spearman’s correlation).

The results, page 6, have been updated:

“Correlation of age and ADCC antibody responses (Fig. 3ghi), showed an age-related increase in pN1 ADCC responses, with older ‘vaccine failure’ children (V1S1) having the lowest pN1 ADCC responses (Fig. 3g, $r=-0.492$, $p=0.0058$), whereas no significant differences by exposure type were observed for pH1 (Fig. 3f) and NP (Fig. 3h) ADCC responses.”

R2.10) Fig. 5e: What point is being made in this panel? Correlation between CD107a and FcgR3a binding has been shown in Fig. 1.

A2.10. This plot has been removed to streamline results.

R2.11) Fig. 6: How long post-infection (where known)?

A2.11. Infection occurred between post vaccination sampling and post seasonal epidemic sampling, which was sampled at an interval of 8 months to 1 year between collections.

A row has been added to table 1 for the month interval for sub study 4 samples, which was V1S1 10.78±1.25 months sampling and V0S1 10.85±1.5 months sampling between vaccination and post epidemic ascertainment of infection.

This is described at the results, page 4:

“Children were monitored for acquisition of influenza infection by serological confirmed infection (S1) by yearly post epidemic sampling for standard haemagglutinin antibody inhibition (HAI) and infection was inferred by >4-fold rise in HAI response, and RT-PCR sampling where influenza like illness was reported, or they remained seronegative (S0) in the following year.”

R2.12) Extended Data 4 legend is incomplete

A2.12. We apologise for this oversight, the extended figure legend has been updated to describe volcano plots.

R2.13) Line 254: Times post infection for the ‘d3’ time point are not given (and in most cases likely not known). Since antibody titers wane rapidly from their peak, comparing the magnitude and diversity of antibody responses at unknown and variable times post infection to a peak time point post vaccination (‘d2’) is likely not very informative.

A2.13. We agree with this reviewer comment and the timing of sampling is limited by practical realities of prospective studies with post season case ascertainment. Therefore, our main comparisons are made at 1 month post vaccination (d2, now t2) between study groups to determine what features may help to reduce acquisition of pandemic infection. Children were not targeted for active blood draws during infection, as infection was inferred by post season sampling, therefore no earlier more proximal blood sample to vaccination is available.

D1 and d2 samples are 1 month apart, and d3 samples are 8 months to 1 year later.

R2.14) Is it possible the generally more robust serological profile of the V0S0 group is because it is partially or wholly comprised of non-seroconverting "infection failures" referred to on line 329 whose antibody breadth and magnitude are being boosted by subclinical infection?

A2.14. There was no seroconversion (by 4-fold rise in HAI responses) in V0S0 group to H3 or influenza B infections, but whether infection occurred without seroconversion or inter-season waning is unable to be determined. The robust serological profile of the V0S0 group could also be attributable to related H1N1 infection in recent years.

We have discussed, that host factors, recent exposure or antigenic gaps may contribute, at page 11:

“Particular immune or genotype features of V0S0 children should be further studied, similar to studies of HIV elite controllers, that enable immune capacity to pre-empt viral diversity.”

and

“further host factors (reviewed in²⁴) such as genotype for key immune pathways (e.g. IFTIM3, CD55, IL28B) or antigenic gaps in prior immunity (no previous or recent H1N1 exposure) should be investigated²⁵.”

R2.15) Line 342: "...these cross-reactive responses are less focussed and not trained by seasonal virus exposure..." If not by exposure to seasonal viral strains, how do these cross-reactive responses arise?

A2.15. We agree the source of V0S0 children's to have both pH1 and H-specific antibody features with FcR effector functions, may enable them to have better outcomes if a divergent influenza virus was encountered. And the source of these responses is unknown. Host factors, such as the ability of Fc glycans to stall B cell responses in germinal center to enable further affinity maturation which has led to greater HA-stalk reactive responses (Wang et al., Cell

2015), is associated with particular IgG1 and IgG3 allotypes. We have added more studies are needed in this area, but without solid evidence of host factors can not speculate further here, and with no available cellular samples from our archive we can not assess this currently.

R2.16) Line 52: "...vaccinated children that became infected (V1S1)...made short term IgG3 with a H3-HA focus post vaccination" is contradicted by line 202 and Fig. 5c, f, and g, in which the V1S0 group has more sH3 IgG3.

A2.16. This comment is based on Figure 5b which shows significantly increased post vaccination responses to a cluster of H3-HA proteins from 1999-2011, and Figure 5e, which uses aggregate HA-specific data (for π), whereby V1S1 post vaccination(d2) IgG3 responses are the highest of any group, but within 1 year (d3) return to levels comparable to d1, and thus not sustained. Whilst figure Fig. 5c, f, and g, in which the V1S0 group has more sH3 IgG3 is one strain. This comment in the abstract has been removed for clarity and we apologise for the conflicting descriptions.

From:

"Whilst vaccinated children that became infected (V1S1) had lower pH1N1 antibody responses than other groups and made short term IgG3 with a H3-HA focus post vaccination, whilst pH1-IgA1 was evident post infection which may circumvent future infections."

To:

"Whilst vaccinated children that became infected had lower pH1N1 antibody responses prior to infection they gained pH1-IgA1 post infection which may circumvent future infections."

R2.17) P-values are missing in some locations in the text (e.g. line 170)

A2.17. Apologies for our oversight, we have checked the manuscript and added p values where appropriate.

For example, this sentence was edited:

From

"There was no difference in baseline sH1 ADCC responses between the 4 groups, vaccinated uninfected (V1S0), vaccinated infected (V1S1), unvaccinated (placebo) uninfected (V0S0) and unvaccinated infected (V0S1) (Fig. 4b), whilst pH1 and pN1 ADCC antibodies (Fig. 4cd) were significantly higher in V0S0 children compared to V0S1. Furthermore, the NP-ADCC response in V0S0 children was also significantly higher than V1S0 children (Fig. 4e)."

To

"Whilst significantly higher responses in V0S0 children compared to V0S1 were found for pH1 ($p=0.0427$) and pN1 ($p=0.0017$) ADCC antibodies (Fig. 3cd). Furthermore, the NP-ADCC response in V0S0 children was also significantly higher than V1S0 children (Fig. 3e, $p=0.0226$).

Reviewer #3 (Remarks to the Author)

The paper describes evaluating serological correlates of protection against pandemic H1N1 2009 illness in children in Hong Kong in archival serum samples. There are only limited studies of correlates of protection in children against influenza and this study provides important results for understanding the complexity of correlates of protection. The use of archival, serum samples collected during placebo RCTs with long-term follow-up for influenza like illness, provides the optimal study design for these evaluations. The work has focused on binding antibodies to seasonal, pandemic and avian, influenza proteins, as well as evaluating more broadly reactive ADCC responses. The cohorts are in principle, large enough to analyse for broad reactivity and the study has three specific aims. The choice of children and methodology are appropriate to meet the study aims. The data in this paper are thorough and the results are interesting. However, unfortunately this paper requires a major revision to improve the readability, understanding and interpretation of the results. In its current form, it is very difficult to

read, as many sentences are trying to say too much, and the key messages are missed by the reader. If the paper is read thoroughly by the authors and rewritten so that a number of sentences are divided to clearly explain the results then the paper will provide important findings for the field.

We expressively appreciate the reviewers substantial comments below which we have addressed to considerably improve our manuscript. We have made edits throughout- including figure and results section titles, figure legends, and removed obstruse sentences as suggested.

Major considerations.

R3.1) The terminology used to describe the groups is difficult and it will be better if the groups are properly described in the text and particularly in the figure legends.

A3.1. We appreciate the reviewers confusion on the 4 groups, and have defined this in the timeline of each figure and in the results section:

“There were 4 possible types of samples used, V1S1 (vaccinated infected), V1S0 (vaccinated uninfected), V0S1 (unvaccinated infected) and V0S0 (unvaccinated uninfected).”

Often in the text in results and discussion we describe both the simplified group names and their vaccination/infection status.

R3.2) it is unclear how many unique children’s samples are involved in this study. I think it would be helpful to include the number of children (n) in figure one under the aims and in the legends of each figure.

A3.2. At each figure panel with a unique sample set the sample size number per group is given in the figure, legend and main text, the groups are also described in Table 1, and the full data from the manuscript are given as a supplementary.

For example at Figure 3a:

Vaccination (V1) Infection (S1)
or Placebo (V0)

post vaccination, pre infection

- V1S0 (n=27)
- V1S1 (n=30)
- V0S0 (n=25)
- V0S1 (n=32)

R3.3) different sources of antigens were purchased, and it is important to include descriptions of whether the proteins are in their native form as HA trimers and NA tetramers and the aa regions of the proteins in the assays should be included in the table.

A3.3. The methods have been updated to state full length HA and NA proteins were not in their native form.

From

“Recombinant proteins (Table 3) were purchased from Sinobiological, MyBiosource, Sigma Aldrich, Merck, and BEI resources, and custom headless HA-trimer ministem proteins for HA-stem proteins (from Raghavan Varadajaran, Indian Institute of Science)³⁵.”

To

“Recombinant full-length monomer-form HA, NA and NP proteins (Table 3) were purchased from Sinobiological, MyBiosource, Sigma Aldrich, Merck, and BEI resources. Custom headless HA-trimer ministem proteins for HA-stem proteins (from Raghavan Varadajaran, Indian Institute of Science)³⁵.”

R3.4) the authors often do not properly explain which children’s groups are being compared, and this needs to be addressed throughout the results.

A3.4. The manuscript has been checked for clarity between comparison groups.

R3.5) What was the likely exposure to pandemic H1N1 in the children? were there school closure or other factors which may have influenced if children were exposed which can help to define correlates of protection

The discussion has been updated:

“While schools were closed in April 2009 for two months to slow the spread of H1N1, ultimately when schools restarted in September 2009 it was estimated that more than 50% of children were infected within a few months, and this is the period covered by our study. Monovalent pandemic H1N1 vaccine became available in January 2010 with a very low community uptake^{1,20}. Therefore, this study aimed to determine the contribution of prospective serological responses that were associated with lack of pandemic H1N1 infection.”

R3.6) unfortunately in its current form the paper is too difficult to read and understand due to the complexities of the sentences.

A3.6. The text has been revised and edited significantly to shorten sentence structure.

R3.7) Clear thought should be given to abbreviations and they should be consistent throughout the paper e.g. Mainly pH1N1 is used but sometimes pdm is used, pdm is the commonly used abbreviation and would make a clear distinction from seasonal H1N1

A3.7. We apologise for inconsistencies, sH1N1 for seasonal and pH1N1 for pandemic. pdmH1N1 has been changed throughout to pH1N1 where appropriate for consistency.

Minor consideration

R3.8) Title: are they a single correlate of protection or correlates of protection?

A3.8. We apologise for the grammatical error, the title has been revised

From:

“Influenza antibody breadth and effector functions are a correlate of protection from pandemic infection of children”

To:

“Influenza antibody breadth and effector functions are immune correlates from acquisition of pandemic infection of children”

R3.9) Abstract: structuring the abstract into introduction methods, results and conclusions may improve the readability. Please include how many children are included in the study, and that long-term follow-up for five years. Please define abbreviations upon first use. I would avoid using the VO, V1, SO, S1 abbreviations in the abstract (and the whole paper) as it unnecessary.

A3.9. The abstract has been significantly modified and abbreviations of groups removed. However, for the sake of figure labelling we have retained the grouping of V1/V0 and S1/S0, but described the group as infected/uninfected, vaccinated or unvaccinated in the text at their first instance.

The group and sample size are given in Table 1, and a total of 391 unique serum samples were used in the study, but group sizes varied depending on the outcome from 30 to 6 children, which is given per figure panel.

R3.10) Line 33 missing a word resulted in partial vaccine effectiveness.

A3.10. Added ‘in’ -> resulted in partial vaccine effectiveness

R3.11) Lines 34-37 perhaps split into two sentences. Also residual protection needs to be explained

A3.11. Changed “which may explain residual protection” to -> “which may contribute to pH1N1 protection attained by seasonal vaccination.”

R3.12) Line 37 several places throughout the paper term immune serum is used, serum is sufficient

A3.12. Updated “immune serum” to just “serum”

R3.13) Line 40 RT PCR is also used to define infection, not only sero conversion

A3.13) Correct, however all RTPCR children also seroconverted, but infections were identified by seroconversion not RTPCR. Therefore, we respectfully retain this brief wording in the abstract. Viral loads that were given in Table 1 have been removed to avoid over emphasis of the limited RTPCR viral load data.

R3.14) Line 43 isotypes should come before subclasses, IgA1 is not an isotype.

A3.14. The abstract has been updated accordingly:

From: “their antibody subclasses (IgG1/2/3) and isotypes (IgG, IgM, IgA1)”

To: “their antibody isotypes (IgG, IgM, IgA) and subclasses (IgG1/2/3).”

R3.15) Line 44 the study only had day 30 and one year follow-up samples, so it would be better to say seasonal vaccination increased H1N1 specific antibodies which declined at 1 year

A3.15. Abstract updated:

From: “We found that seasonal vaccination briefly increased pH1N1 specific antibodies and FcR effector functions, but they declined within one year post vaccination.”

To: “We found that seasonal vaccination increased pH1N1 specific antibodies and FcR effector functions, which declined within one year post vaccination.”

R3.16) Line 48 which other groups?

A3.16. ‘Other groups’ referred to : unvaccinated infected, vaccinated infected and vaccinated uninfected, but to describe 3 comparison groups is long winded. ‘Other groups’ has therefore been removed.

R3.17) line 50 in their antibody responses instead of to

A3.17. Changed ‘to’ to ‘in’

R3.18) Line 52 which other groups? Also short term you did not have samples between day 30 and 1 year

A3.18. Removed short-term

R3.19) Line 55 HAI remove antibody as the definition is incorrect

A3.19. Updated to “hemagglutinin inhibition antibody responses”

R3.20) Line 57 what are FcR binding features?

A3.20. Updated text to move features -> “in combination with features that include pH1-IgG1, H1-stem responses and FcR binding”.

R3.21) Line 60 include H1N1

A3.21. added pH1N1 infection.

R3.22) Line 65 this is incorrect elderly and people with comorbidities experience the highest hospitalisations.

A3.22. Sentence updated

“Children are particularly susceptible to influenza virus infection and represent the highest number of hospital admission during seasonal epidemics, whilst individuals with comorbidities and older adults have higher mortality.

R3.23) Line 66 can you rephrase this e.g. are not designed to protect

A3.23. Updated sentence to “Seasonal influenza vaccines have limited protection against pandemic virus infection”

R3.24) Line 69 divide into two sentences,

A3.24. Updated

R3.25) Line 75 against pandemic infection, this sentence does not make sense and again should probably be split into two sentences.

A3.25. Sentence deleted.

R3.26) Line 77 perhaps change to limiting infection (instead of acquisition) and severity of disease?

A3.26. “limit infection” now used.

R3.27) Line 84 subtypes and strains (in order of diversity)

A3.27. ‘even subtypes’ added.

R3.28) Line 85 perhaps include influenza A H1N1, H5N1 and H7N9.

A3.28. Added influenza A viruses following.

“can be highly cross-reactive for different strains and even subtypes of influenza, whereby ADCC antibodies towards the pandemic H1N1 (pH1N1), highly pathogenic avian H7N9 and H5N1 influenza A viruses”

R3.29) Line 85 Lethal- they are commonly referred to as highly pathogenic

A3.29. Updated to ‘pathogenic’

R3.30) Should the structure of stem and division into two groups for Influenza A viruses be explained in the introduction?

A3.30. We have updated this section to include relevant references to describe HA stem and NP directed ADCC, and HA group phylogeny:

“Therefore, cross-reactive ADCC antibodies may target conserved viral regions of the HA, such as the HA-stem⁷ or internal proteins such as the Nucleoprotein (NP)⁸ and Matrix 1 (M1), facilitating FcR binding antibodies to have greater subtype cross-reactivity than conventional neutralizing antibodies^{5,8}. Indeed, influenza A viruses can be phylogenetically distinguished based on HA groups 1 and 2, which have distinct HA-stems⁹.”

R3.31) pH1N1 or pdm (see 7 above), pdm would make a clear distinction from seasonal H1N1

A3.31 pH1N1 now used throughout.

R3.32) Line 91 abbreviate Matrix protein which is used later.

A3.32 Abbreviated to (M1)

R3.33) Line 91 Lending should this be allowing. Is a word missing? To provide greater subtype...

A3.33. Updated to “facilitating FcR binding antibodies to have greater”

R3.34) Line 96 is the half-life of human IgG3 only 1 week?

A3.34. Half-life specifications of IgG3 removed.

R3.35) Lines 93-98 divide into at least 2 clear understandable sentences. What does potency with time since exposure mean?

A3.35. Removed 'for potency'

R3.36) Line 101 remove immune

A3.36. Removed immune

R3.37) Line 103 to assess influenza specific antibody breath.

A3.37. Added

R3.38) Line 103 -108 divide into short sentences that explain the study background.

A3.38. Paragraph divided for sentence structure, and the RCT study background is described in more details at the first results section.

R3.39) Line 109 change to children were monitored for influenza H1N1 infection. Define RT-PCR.

A3.39. Updated to "Where influenza like illness was reported, daily symptom diaries were recorded and nasal swabs were collected for real time polymerase chain reaction (RT-PCR) sampling to determine viral loads."

R3.40) Line 110 change to serologically

A3.40. Updated to "Children were monitored for serological confirmed acquisition of influenza infection (S1) by yearly post epidemic sampling for standard haemagglutinin antibody inhibition (HAI) and infection was inferred by >4-fold rise in HAI response, or inferred as uninfected children who remained seronegative (S0)."

R3.41) Line 112 RCT is now described as a cohort study

A3.41. Corrected to RCT.

R3.42) Line 114-5 include avian H5 and H7 viruses

A3.42. Section removed.

R3.43) Line 116 describe 3 aims in introduction.

A3.43. Added to introduction and results: "This current study selected samples to address research gaps in antibody effector functions: (i) to determine vaccine induced ADCC changes, (ii) longitudinal stability of vaccine induced antibody FcR binding and isotype changes, (iii) differences baseline FcR binding and isotype with acquisition of pH1N1 infection, and (iv) the diversification of HA-specific antibody responses with vaccination and infection."

R3.44) Line 117 delete to be made for

A3.44. Section removed.

R3.45) lines 119-122 separate in short clear sentences explaining the aims. Here pandemic H1 N1 infection is described as pdmH1N1, not pH1N1.

A3.45. pH1N1 corrected.

Updated section to :

"Previous exposure to seasonal H1N1 viruses in older adults was attributed to reduced severity of pH1N1², and there is an age-related accumulation of ADCC responses⁵, therefore we aimed to determine if ADCC responses were maintained long term post vaccination. To assess the longitudinal stability (versus initial year 1 day 0 baseline) of seasonal vaccine stimulated antibody responses (IgG, IgG1, IgG2, IgG3, IgA, FcγR3a and FcγR2a responses for seasonal and pH1N1 representative proteins were used: H1 (sH1), pH1, sH3, H1-stem, pN1, sN1, M1 and NP proteins. Children were also selected based on the criteria of no further vaccination or infection (by seroconversion) during the subsequent 5 years (Fig. 2), and thus we could measure the stability of vaccine responses alone compared to placebo controls."

R3.46) Line 125 remove one of cell in title. Again this is heading is trying to say too much in one sentence. Focus on the important findings.

A3.47. Results section title changed

From: "Protected children have higher baseline pandemic-specific ADCC responses and diverse influenza virus recognition"

To: "Children who do not acquire pH1N1 infection have higher baseline pandemic-specific ADCC responses"

R3.47) Line 127 a brief introduction to the samples and the study aims would be a improve the start of the results section.

A3.47. Added: "To assess prospectively for antibody correlates prior to infection, we selected children known seroconvert (S1) to pH1N1 in the next year or not (S0), who were initially randomised for vaccination. These prospective baseline samples were assessed in a NK cellular function assay for ADCC responses against pH1N1 and seasonal vaccine representative proteins (Fig. 3).

R3.48) Lines 128-132 again, please split the sentence into the methods and then the findings. And the correlation show binding antibodies versus degranulation.

A3.48. Correlation removed.

R3.49) Figure 1a I think this figure needs more thought to clearly explain the study set up

A3.49. Figure 1a removed, and each figure has their own sample selection timeline.

R3.50) please include the number of children in each aim and that the follow up samples are collected yearly for 5 years.

A3.50. Added.

R3.51) Aim 1.1 does not make sense. Seasonal vaccination for induction of pandemic ADCC.

A3.51. From introduction and results section

"(i) to determine vaccine induced ADCC changes (Fig. 1)"

R3.52) Aim 1.2 longitudinal stability of what?

A3.52. From introduction and results section

(ii) longitudinal stability of vaccine induced antibody FcR binding and isotype changes (Fig. 2),

R3.53) Could aims 1.1 and 1.2 be combined into one aim? e.g. Impact of seasonal vaccination on Pandemic ADCC and durability

A3.53. A larger sample size was available to assess vaccine induced changes to the antibody effector functions (n=30) for Aim 1.1, whilst due to exclusion of further infection or vaccination a smaller sample size was available for A1.2 (n=6/8). Therefore, data is not combined.

R3.54) Aim 3 should this be HA diversity of antibody responses

A3.54. From introduction and results section

"(iv) the diversification of HA-specific antibody responses with vaccination and infection"

R3.55) There are a lot of undefined abbreviations in the figure legend.

A3.55. Figure legend significantly updated, MFI defined, and statistical tests updated.

R3.56) Can you clearly state blood samples were collected baseline (day 0), day 30 (post vaccination) at 1 year follow up in 2010 and for a subgroup (n=x) yearly between 2010 and 2014.

A3.56. Added.

“To assess the longitudinal stability (versus day 0 baseline, day 30 (post vaccination, 2009) at 1 year follow up in 2010, 2011, 2012, 2013, 2014)”.

R3.57) Think about the use of brackets in the figure legend

A3.57. Checked.

R3.58) Delete proteins of interest

A3.57. Deleted.

R3.59) Line 134 divide sentence into two, e.g. although postvaccination a significant increase in fold change for pN1 responses was observed compared to placebo only. (you can say, placebo only and delete controls)

A3.57. Figure removed.

R3.60) Figure 2c is the original data presented in the paper or just fold change. It appears there only six individuals who have higher responses. Have you conducted a sensitivity analysis to show if these individuals are unduly influencing the results?

A3.60. Figure removed.

R3.61) Lines 137-140 can you simplify this Having confirmed the correlation between multiple binding to NK cellular functional degranulation, multiplex assays were used to evaluate multiple responses which is ideal for paediatric samples as it only requires low serum volumes.

A3.61. Added.

R3.62) Line 142 is durability a better word than longitudinal stability? Should this read after seasonal vaccination instead of by vaccination? Perhaps start with this is a small subgroup of 14 subjects.

A3.62. Updated to ‘durability’ in results section title and seasonal removed.

R3.63) Table 1 is the use of mean age suitable for children, often median age is used in paediatrics

A3.63. Both the mean with SD and range is given for clarity. The median has been added to Table 1.

R3.64) Line 144-145 move to discussion as not relevant for children

A3.64. Moved from results to discussion:

“Antibody waning is a natural feature of plasmablast B cell response contractions to form a stable B cell memory pool, and the half-life of different antibody subclasses varies substantially. Previous exposure to seasonal H1N1 viruses in older adults was attributed to reduced severity of pH1N1², and there is an age-related accumulation of ADCC responses⁵,

R3.65) Lines 146-151 divide into shorter sentences

A3.65. Shortened to 3 sentences.

R3.66) Line 149 Matrix protein has not been previously defined

A3.61. Now defined in introduction.

R3.67) Lines 151-156 this is incomprehensible sentence, please split into shorter sentences

A3.67. Sentence deleted.

R3.68) Lines 162-166 perhaps shorter sentences and define the four groups at the start

A3.68. Defined at results section

“There were 4 possible types of samples used, V1S1 (vaccinated infected), V1S0 (vaccinated uninfected), V0S1 (unvaccinated infected) and V0S0 (unvaccinated uninfected).”

R3.69) Figure 2

A3.39. Figure 2 has been merged to figure 1, some panels remain.

- a. **It would be helpful to have placebo and vaccination as subheadings in the figure legend**
The figure legend describes: Serum paired samples for pre (day 0) versus post (day 30) vaccination (V1) or placebo controls (V0),
- b. **be consistent in the terminology used to describe the NK cell ADCC**
All graph labels use “%CD017a+ of NK92 cells”
- c. **c I would include placebo and vaccination on x axis**
X axis labelled for V1 (vaccinated) and V0 (placebo), and defined in Figure 1a.
- d. **Figure 2c where is the original data. It appears there only six vaccinated individuals who have higher responses. Have you conducted a sensitivity analysis to show if these individuals are unduly influencing the results?**
Figure removed.
- e. **There are a lot of undefined abbreviations in the figure legend.**
Proteins are defined in the text, and MFI now defined.
- f. **Think about the use of brackets in the figure legend**
Checked

R3.70) Figure 3a can you change to show yearly samples between 2010- 14

A3.70. Individual yearly data as a fold change from pre vaccination is shown.

R3.71) Figure 3 The ordering of graphs is strange should NP (b) be beside M1 and stem together with H1

A3.71. The order follows the antibody detector for IgG, IgG1, IgG3 and FcgR2a.

R3.72) Table 1 the symptom score should be defined in the legend and do you need to separate this into two lines of the table? If so clearly define each as refereeing to either RT-PCR or serologically confirmed infection.

A3.72. Data is very limited for RTPCR titers, and all infections were confirmed by seroconversion. The viral load and symptom details of table 1 have been removed for clarity as this does not represent how infections were identified.

R3.73) Can children be in several groups or are they only in one group? Table 2 and Table 3 do these need to be a main tables or can abbreviations be included into the figure legends?

A3.73. Due to limited sample volume children infected children were used once.
Table 2 and 3 are now extended data table 1 and 2

R3.74) Lines 174-177 the rationale for using HAI titre of 40 as detectable titres needs to be included as detectable titres are often 10 or 20 (depending upon the stating dilution), whilst 40 is considered protective in adults. Some of the text from the paragraph on HI titres for H3N2 in the discussion could be used. Normally HAI are presented as GMT for each group.

A3.74. For clarity HAI responses are now shown per group as the first figure.
A discussion includes:

“Children need a higher level of HAI antibodies estimated at 1:320 for 80% protection against infection¹⁸, compared to the WHO accepted standard of 1:36 from adults for 50% protection¹⁹, which only provides 22% protection in children.”

R3.75) Line 183 yo is missing after 9-11

A3.75. Age stratification is now updated to continuous for analysis.

R3.76) Line 188 can you write this more clearly e.g. The Multiplex assay was expanded to measure both binding (isotype/subclass) and functional binding antibodies against pandemic, seasonal and cross reactive proteins.

A3.75. Added.

R3.77) Line 192 change to simplify however, no differences....

A3.75. Sentence restructured to:

“Whilst NP-IgG1 was elevated in vaccinated children (Fig. 4b), there was no difference in NP-specific IgG3 (Fig. 4c) or Fc γ R3a binding (Fig. 4d), despite our earlier finding that V0S0 children had increased NP-specific ADCC cellular responses (Fig. 3e).”

R3.78) Line 194 is this correlation with the same children?

A3.78. To clarify the correlation in Figure 1ef is not the same V0S0 children, but pooled from all groups of Figure 3. The figure 1ef legend has been updated, and reference to the correlation between NP-specific ADCC NK92 cell function and Fc γ R3a binding removed.

R3.79) Line 196 change per to for

A3.79. As this refers to the fold change of Fc γ R3a to IgG3, we have changed ‘per’ to ‘to’.

R3.80) Line 197 which is befitting perhaps change to which is as expected.

A3.80. Updated to ‘as expected’.

R3.81) Line 200 can you simply say Multiplex comparisons of post vaccination

A3.81. Updated to:

“A heatmap overview, of z-scored log transformed data, antibody features detected by multiplex (Fig. 4f),”

R3.82) Line 204 the sentence refers to studies but only reference 14 is provided (also is this a discussion point)

A3.82. This sentence has been removed from the results, and the New Zealand influenza B virus and Nicaraguan H3N2 association with H1N1 infections are discussed.

R3.83) Line 204 to 205 simplify to Whilst comparison of infected and uninfected vaccinated children

A3.83. Updated.

R3.84) Line 207 simply put the p value next to the significant values.

A3.84. Sentence and figure deleted.

R3.85) Line 220 how many samples were paired? Include n

A3.85. Removed ‘paired where available’, as the group is compared here and the available samples is dependent on donor and timepoint. For example, for the V1S1 group there are 16 participants, only 2 participants have samples from 3 timepoints, 4 have pre/post infection paired, whilst 4 have pre vaccination/post infection and 1 has pre/post vaccination, and remaining have single timepoints samples. This would be complicated to describe and fold changes are not shown per donor in this analysis.

R3.86) Line 227-229 rewrite the sentence as it is difficult to understand

A3.86. The description of these results has been updated to:

“To break down antibody specificities for significant differences from the heat map, the mean difference between groups and detectors of interest were visualised by volcano plot analysis (Extended data 4). Vaccinated infected children, V1S1, had minimal differences to V1S0 children for IgG1 responses, with only seas. H1-1999 being significantly elevated ($q=0.04$) (Extended data 4a).”

R3.87) Line 237 why is d2 timepoint introduced here and can you simply give the days post vaccination. This applies to all subsequent descriptions. I think you are trying to say H3 antibody responses correlates with Igs and FcR, but this was not observed for other parameters.

A3.87. We agree and the figure 5e and Table 1 has been updated with labels as day 0, day 30 and 1 year.

The description of the PCA (Fig. 5c), has been updated:

From

“A combined PCA of post vaccination pre infection samples, i.e. ‘t2’ timepoint, (Fig. 4c), which captured total variance of the data (Dim. 1: 24% and Dim. 2: 18%), showed that seasonal H3-HA antibody responses corresponded between Ig’s and FcR binding, whilst H1-HA, influenza B and avian (H5 and H9) antibody responses PC was not likely to be correlated to H3-HA responses.”

To

“A combined PCA of post vaccination pre infection samples, i.e. ‘t2’ timepoint, (Fig. 4c), which captured total variance of the data (Dim. 1: 24% and Dim. 2: 18%), showed that H1-HA, influenza B and avian (H5 and H9) antibody responses PC was not likely to be correlated to H3-HA responses.”

R3.88) Line 254 d1 versus d2 has not previously been defined, perhaps use pre-to post

A3.88. We agree and the figure 5e and Table 1 has been updated with labels as day 0, day 30 and 1 year.

R3.89) Discussion: The discussion could do with being better structured lines, focusing first on the main findings, moving the limitations to the second from last paragraph and ending with conclusions.

A3.89. We agree, and the discussion has been streamlined, and limitations moved to the second last paragraph.

R3.90) Lines 288 to 304 are more background. Perhaps the section on HAI antibodies should be moved to the introduction

A3.90. We appreciate the reviewers keen edits, and have moved this description of HAI, FcR and T cell responses to the introduction as a more appropriate place.

R3.91) Lines 318-319 move specific antibodies to after the FC region, change to due to limited archival sample volume

A3.91. Updated

R3.92) Lines 331-335 could be maybe moved to the start of the discussion at the end of the first paragraph

A3.92. Moved as suggested, again thank you for your thoughtful edits.

R3.93) Line 349 delete were then

A3.93. Deleted

R3.94) In children protective IFN- γ T cells have been found after LAIV and should be mentioned in the discussion (PMID: 18448618)

A3.94. Added to discussion of immune correlates, page 11.

“Whilst cross-reactive memory T cells are reported to contribute to protection from symptomatic infection²⁴, and can be boosted in children by live attenuated influenza vaccines²⁵.”

R3.95) Data availability, the data should be supplied as deidentified for each figure/table. Or at the very least the contact for the data should be included

A3.95. As per Nature Communications policy, and our prior publications with the journal, we intend to include the raw data of each figure panel as supplemental data. At the time of submission and subsequent substantial changes made during revision, the raw data needed

further coordination to align with figure panels. The data availability statement will be updated to 'attached' before final publication. In the meantime, we have updated our statement to "Deidentified raw data that support this study are available by request from the corresponding author.

Methods

R3.96) Line 386 was vaccination conducted in 2008 or 2009, as the vaccine used was the 2009-10 TIV

A3.96. We apologise for this oversight, the vaccine year has been updated from 2009-2010 to 2008-2009 and 2009-2010, as samples are drawn from both these studies.

R3.97) line 392 delete one of based

A3.97. Deleted

R3.98) line 397 did both parents provide informed consent or only 1?

A3.98. One parent gave consent, and the child gave assent. In most cases both parents were not present at the time of recruitment.

R3.99) Line, 433 incubate should this be in the past tense? Do you mean sera?

A3.99. Updated to incubated.

R3.100) The HAI methodology is missing. Why have the authors chosen to use a titre of 1:40 as positivity? Is this because it has been associated with protection in adults?

A3.100. We refer to published HAI data from this study. A 4-fold rise in HAI status was used to identify pH1N1 infections. The seroprotective rates are discussed in the introduction and a titer of 320 is needed in children for equivalent protection to adults titer of 36.

R3.101) Line 493 Missing upon log transformed

A3.101. Updated, "upon log transformation and the data were further"

R3.102) Line 498 change to we measured

A3.102. Updated.

Reviewer #4 (Remarks to the Author):

This manuscript reports on an analysis of immunological correlates of protection from influenza infection, using samples from a historical placebo-controlled trial of seasonal influenza vaccination.

Seasonal influenza vaccination was found to increase pH1N1 specific antibodies and FcR effector functions, which declined over 1 year post-vaccination. Antibodies and FcR effector functions post-vaccination were not clear correlates of influenza infection among the vaccinated, although pH1 and pN1 ADCC antibodies were higher in magnitude among unvaccinated uninfected vs. infected individuals.

The statistical methodology is difficult to follow, and in some instances overlooks issues of confounding and multiple comparisons and does not address the stated objectives of the study. In particular,

R4.1) basic descriptive statistics are missing, eg how many cases and controls were tested at each time point, what is the overall level of efficacy of the seasonal vaccine, how many cases and controls in training and test sets, how many subjects for addressing each aim

A4.1. To clarify, the sample size is given per figure panel and overall in Table 1. Figure legends include statistical tests used, which include adjusting for multiple comparisons. The seasonal vaccine offered 47% vaccine efficacy in this trial, as reported in Cowling *et al.*, CID 2009.

Which is described in the first paragraph of the introduction. The full data set for per participants timepoints are given in the raw data associated with the manuscript.

The parameters of the logistics regression model are described in the methods, including parameters, and samples for testing and training. (see page 19 'Logistic regression').

"Logistic regression models for prediction of infection were built using sub study (iii) data (n=115 samples) as training set and sub study (iv) data (n=36 samples) as validation sets. We included 4 proteins (i.e., sH1-2007, pH1, sH3-2007 and H1-stem) and 6 antibody features (i.e., IgG, IgG1, IgG3, IgA1, Fc γ R3a, and Fc γ R2a) (Table 2), resulting in the comparison of 24 variables. The input variable of the predictive model are the 24 antibody responses at post vaccination pre infection timepoints, the output value is whether they become infected (S1 or S0 group, infection as 1 and non-infection as 0). Models were built using R scripts which can be accessed via https://github.com/Leo-Poon-Lab/analysis-Influenza-antibody-subclass-and-effector-functions/blob/main/scripts/Predictive_model.R.

The formula to calculate accuracy is:

Accuracy = (True Positives + True Negatives) / (True Positives + True Negatives + False Positives + False Negatives)

The formula to calculate sensitivity is:

Sensitivity = True Positives / (True Positives + False Negatives)

R4.2) Objectives are not laid out clearly enough (Figure 1a).

A4.2. This figure overview has been deleted for clarity. And the per sub study purpose is given in the introduction and results, and the samples used described in table 1.

For clarity the text below appears in the introduction and results:

"In this study, we aimed to assess influenza specific antibody breadth and function between sH1N1 vaccination and pH1N1 infection using archived samples from a randomized placebo control trial (RCT) of seasonal inactivated influenza vaccines in school aged children that were collected at the onset of the 2009 H1N1 pandemic. We measured antibody responses by their isotype and subclass, Fc γ R3a and Fc γ R2a binding against diverse HA, NA, or conserved NP and M1 proteins before and after seasonal vaccination, over time, and during the 2009 pandemic for acquisition of pH1N1 infection. We also explore features of antibodies that may correlate with reduced pH1N1 infection. This current study selected samples to address research gaps in antibody effector functions: (i) to determine vaccine induced ADCC changes, (ii) longitudinal durability of vaccine induced antibody FcR binding and isotype changes, (iii) differences in baseline FcR binding and isotype with acquisition of pH1N1 infection, and (iv) the diversification of HA-specific antibody responses with vaccination and infection."

R4.3) Lines 160 and following, where the immune correlates analysis is reported, unclear why a Kruskal-Wallis test is used as the contrasts are between two groups

A4.3. For figure 3, "Comparisons between groups were performed using Kruskal Wallis with Dunns multiple comparisons test", as 4 vaccine/infection groups are included.

R4.4) Figure 3, unclear if there is adjustment for the multiple comparisons that are done, and why there are tests at each time point rather than tests for trends over time which would be more powerful.

A4.4. Now Figure 2, A mixed effects model has been used instead for comparison of longitudinal data, which identified further waning from the unvaccinated group across numerous parameters. This was also raised and addressed at comment R2.7.

The figure 2 legend has been updated for statistical tests used to exclude outliers and mixed linear models when comparing vaccinated (V1) to unvaccinated (V0) children:

"Outliers were excluded using ROUT identification at Q=1% leading to exclusion of n=1 in V0 NP IgG, n=2 in V0 H1stem IgG, n=1 in V1 M1 IgG, n=1 V0 and n=1 V1 pH1 IgG1, and n=6 V0 and n=1 V1 pH1 IgG3. Statistically significant differences shown above (by mixed linear model with Tukey's test correcting for multiple comparisons, between years* coloured for comparison group V1 blue, V0 red) and in tables below versus V0 group (by mixed linear model with Tukey's test correcting for multiple comparisons) or theoretical value of 1 (for no fold change, by one sample Wilcoxon test)."

R4.5) Figures 4 and 5: some of the contrasts are uninterpretable. Eg why contrast

vaccinated infected individuals (V1S1) with unvaccinated uninfected individuals (V0S0)? The objective here is to understand predictors of infection, which involves the contrast between V1S0 and V1S1, and between V0S0 and V0S1.

A4.5. We agree and the comparison of V1S1 vs V0S0 has been deleted, which intended to show that trends were similar for the V1S0 group, however this has been removed for clarity.

R4.6) It is not clear if any covariate adjustment was done when contrasting infected vs. uninfected individuals, beyond the matching for age. A rigorous analysis would include adjustment for potential predictors of immune response and risk of infection, to control for confounding.

A4.6. In the present study, only different antibody responses (and no other covariates) were included in models for prediction of infection. We agree there could be potential confounding in the present model, and therefore we have included the following statement in the revised manuscript:

“While selection of infected and uninfected individuals were matched by age and sex, comparison of their pre-infection antibody responses were conducted through prediction models based on machine learning and only included different antibody responses as input variables, thus interpretation of the importance of the antibody responses identified to be predictive in the present study should be cautious due to potential confounding by other factors (such as social contact pattern) that were unadjusted for in our models.”

R4.7) Description of the predictive model evaluation on lines 269-73 is especially difficult to parse; "accuracy" and "sensitivity" metrics are not defined

A4.8. To clarify, the accuracy and sensitivity of the model are described in Extended data 5c. We have added formulas to the methods.

Accuracy: Accuracy is the proportion of correct predictions (both true positives and true negatives) made by the classification model out of the total number of instances in the dataset. It measures the overall performance of the model in predicting the correct class labels.

The formula to calculate accuracy is:

$$\text{Accuracy} = (\text{True Positives} + \text{True Negatives}) / (\text{True Positives} + \text{True Negatives} + \text{False Positives} + \text{False Negatives})$$

Accuracy is a useful metric when the class distribution is relatively balanced, and the costs of false positives and false negatives are similar. However, it may be misleading when the dataset has imbalanced class distribution, as the model may achieve high accuracy by just predicting the majority class.

Sensitivity: Sensitivity, also known as recall or true positive rate (TPR), is the proportion of true positives (actual positive instances that have been correctly identified by the model) out of the total actual positive instances. Sensitivity measures the ability of the model to identify true positive cases among the instances that are actually positive.

The formula to calculate sensitivity is:

$$\text{Sensitivity} = \text{True Positives} / (\text{True Positives} + \text{False Negatives})$$

Sensitivity is particularly important when dealing with cases where the cost of false negatives is high, such as in medical diagnosis or fraud detection. A model with high sensitivity can effectively identify the positive cases, minimizing the false negative rate.

The high accuracy and sensitivity of our model on the training dataset suggest successful capture of the most important information in the data for predicting infection, however the poor accuracy and sensitivity on the prediction task (using data from sub study (iv)) suggest low generalizability of the model and heterogeneity of data from different cohorts.

R4.8) I do not know how to interpret the analyses shown in Figure 6

A4.8. Now Figure 5, presents the sampling timeline, heatmap overview, PCA, Elastic network analysis and HA diversity analysis. This is all based on log transformed z-scored data as

measured by the multiplex platform from samples defined in Table 1, pre/post vaccination or infection based on grouping.

Added to results:

“Correlation network analysis with Feature selection by Elastic Net of selected variables (Fig. 5d, Extended data 4e), showed pH1 antibody features clustered with avian H5-specific IgG and Fc γ R2a and Fc γ R3a, separate from seasonal H1-HA's or HA-stem responses. Red labels correspond to the important variables selected by elastic net with frequency >70% of the total 2,000 iterations, as the important variables in terms of predicting infection”

In response to comment R1.4, we had also added to the results :

From our heatmap and PCA analysis patterns were apparent in the breadth of HA-specific antibody responses by vaccination and infection between groups. Therefore, a 2-D representation was created of responses of different groups and timepoints to compress for HA cross-reactivity (measured by π) versus V0S1 pre-infection samples (day 30 post vaccination timepoint, selected as the most susceptible group to infection) (Fig. 5e). The y-axis represents the diversity of the HA response, and the x-axis represents the magnitude difference from V0S1 day 30 of other timepoints or groups responses.”

R4.9) The small size of the study limits the strength of the conclusions. The longitudinal stability of the antibody responses is especially imprecise given just 14 participants.

A4.9. We agree, hence the initial analysis of 30 vaccinated and unvaccinated children. However due to the strict selection criteria of no further vaccination or infection for 5 years our resulting sample size was 6 and 8 children per group. We include the caveat that strong conclusions cannot be drawn from this dataset.

As described at page 6 of results:

“Children were also selected based on the criteria of no further vaccination or infection (by seroconversion) during the subsequent 5 years (Fig. 2), resulting in a limited sample size of V1 (n=8) and V0 (n=6) due to stringent criteria, which precludes strong conclusions in antibody durability”

More minor comments:

R4.10) The design of the study needs to be more clearly described prior to presenting the results; the description on lines 112-116 is inadequate. Eg stating that a matched case-control study was used, and how many cases and controls were selected and tested at each time point.

A4.10. The methods has a new added section for sample selection (page 14/15), a new first section of results describes the study sample selection (page 5), and introduction has an added paragraph to describe the aims of the study (page 4).

The sub-study sample sizes are described in Table 1, each legend and in each figure when introducing the sampling timepoints.

R4.11) "Baseline" term is confusing as this refers to Day 30, not Day 0

A4.11. We apologise for the confusion, baseline refers to pre pH1N1 infection exposure, and the sample timeline is given per figure to clarify the timepoints being compared.

For longitudinal responses (Fig. 2) are compared to the pre vaccination baseline response, day 0. For later analyses baseline, (Fig. 3 and 4) refers to post vaccination, pre infection, as baseline to infection for the season when infection occurred. We have clarified in the text the timepoint used.

R4.12) "Experiment was repeated twice" is unclear: were there duplicate samples?

A4.12. As stated in the figure legends, duplicate samples were run for the multiplex assay, and experiments repeated twice for the NK92 cellular assay.

Reviewer #1 (Remarks to the Author):

The authors have done a nice job with rewriting and reorganizing this manuscript. Overall it is easier to follow the description of cohorts. The results are caveated appropriately and overall the manuscript is improved from initial submission.

Reviewer #2 (Remarks to the Author):

The authors have adequately addressed my concerns

Reviewer #3 (Remarks to the Author):

The authors have put considerable work into the revision of the paper which has substantially improved the readability and understandability of the paper. The main area which could be improved is the description of the cohort at the start of the results section so that it focuses specifically on what is relevant to this paper. Other information on the study can be referenced.

Reviewer #3 (Remarks on code availability):

The link did not work for me and needs to be checked

Reviewer #4 (Remarks to the Author):

This is a revision of a manuscript aiming to characterize the breadth and effector immune functions induced by influenza vaccination, and their correlation with risk of influenza acquisition.

The main issue with the manuscript is organization, writing, and clarity of objectives, methods, and results. The revision is improved, but I still had considerable challenges understanding the work. I would recommend that the authors work with a scientific writer to present the methods, results, and implications clearly and succinctly. There are also limitations of the statistical analysis that warrant further attention.

Examples of lack of clarity in methods:

The parent study design is still not clear. Was randomization to vaccine or placebo at an individual or household level? Are any of the children sampled for immunogenicity assessment in the same household?

The case-control sampling design is not clear. In some places, it sounds as if matching was done, but in the Methods section matching is not mentioned. The sentences in lines 492-5 are not precise, eg are these separate sets of participants? How were participants lost to follow-up handled? "Selected from the same year" -- of enrollment or ?

Examples of organizational issues:

Section beginning "Children who do not acquire pH1N1 infection have higher baseline pandemic-specific ADCC responses" deviates from this objective and contrasts vaccinated and unvaccinated participants, looks at age vs ADCC responses, and more.

Several results sections have discussion and limitations mixed in, which makes the results hard to follow.

Examples of lack of clarity in results: Lines 188-194 describing longitudinal data analysis is too vague to be of any use.

Examples of issues with statistical methods:

Dunnnett's test is appropriate for pairwise comparisons among groups following an overall result suggesting at least one difference within the set, but it is mentioned in the context of only two groups (eg vaccine vs placebo or pre- vs post-vaccination), so seems inappropriate.

Correlates of risk tests seem to be done both using simple two-group comparisons (infected vs. uninfected vaccinees) as well as using logistic regression. Why the former is used at all is not clear, as the regression model is more powerful in that covariates and other immune markers can be adjusted.

The logistic models fit are massively overparameterized: one model has 24 variables and is fit using 115 observations. A general rule of thumb is no fewer than 10 observations per parameter, which would mean that 250 observations would be needed to fit such a model.

Removing outliers is generally not good practice, unless there is strong evidence that there are actual data entry or assay failure; observing unusually high or low values don't provide sufficient rationale for exclusion

REVIEWER COMMENTS

Reviewer #1 (Remarks to the Author):

The authors have done a nice job with rewriting and reorganizing this manuscript. Overall it is easier to follow the description of cohorts. The results are caveated appropriately and overall the manuscript is improved from initial submission.

Reviewer #2 (Remarks to the Author):

The authors have adequately addressed my concerns

Reviewer #3 (Remarks to the Author):

The authors have put considerable work into the revision of the paper which has substantially improved the readability and understandability of the paper. The main area which could be improved is the description of the cohort at the start of the results section so that it focuses specifically on what is relevant to this paper. Other information on the study can be referenced.

Reviewer #3 (Remarks on code availability):

R3.1: The link did not work for me and needs to be checked

A3.1: Analysis scripts have been deposited at Zenodo under accession code <https://zenodo.org/records/10583684>. This has now been updated in the code availability section.

Reviewer #4 (Remarks to the Author):

This is a revision of a manuscript aiming to characterize the breadth and effector immune functions induced by influenza vaccination, and their correlation with risk of influenza acquisition.

The main issue with the manuscript is organization, writing, and clarity of objectives, methods, and results. The revision is improved, but I still had considerable challenges understanding the work. I would recommend that the authors work with a scientific writer to present the methods, results, and implications clearly and succinctly. There are also limitations of the statistical analysis that warrant further attention.

Examples of lack of clarity in methods:

R4.1: The parent study design is still not clear. Was randomization to vaccine or placebo at an individual or household level? Are any of the children sampled for immunogenicity assessment in the same household?

The study design and primary results of the original randomised vaccine trial which aimed to look at both direct and indirect benefits of vaccinating children in households have been reported (Cowling et al Clin Infect Dis 2012, doi: 10.1093/cid/cis518; Tsang et al Nat Commun 2019, doi: [10.1038/s41467-018-08036-6](https://doi.org/10.1038/s41467-018-08036-6)). Briefly, only households with at least one child aged 6-17 years were enrolled, and one eligible child from each household was randomised to receive a trivalent influenza vaccine (TIV) or a placebo. Serum samples were collected from both children receiving the randomised study vaccination, and their household members. This is to allow assessment of direct benefit of vaccination in vaccinated children, as well as indirect benefit of vaccination to family members due to reduced risk of influenza transmission from vaccinated children. For the present investigation, only one child (who received the randomised vaccination) from the same household was included for each analysis. We have added this information in the revised manuscript.

R4.2: The case-control sampling design is not clear. In some places, it sounds as if matching was done, but in the Methods section matching is not mentioned. The

sentences in lines 492-5 are not precise, eg are these separate sets of participants? How were participants lost to follow-up handled? "Selected from the same year" -- of enrollment or ?

In the revised manuscript we have now provided a detailed description of the process of sample selection, extracted below:

“For the present study, we selected subsets of children 6-17 years of age who had received any influenza vaccination in Year 1 (“vaccinated”/ “V1”) or not (“unvaccinated”/ “V0”) for secondary analyses to study the effect of vaccination on short-term and long-term serologic responses in children who had (“infected”/ “S1”) or had not (“uninfected”/ “S0”) pH1N1 infection in Year 1 (Table 1). In sub study (i), to study the effect of vaccination on cellular ADCC, we randomly selected 10 children from three age groups (≤ 8 years, 9-11 years, ≥ 12 years old) in both vaccinated and unvaccinated groups, and tested their pre- and post-vaccination sera. In sub study (ii), to study the longitudinal stability of vaccine antibodies, separately we randomly selected 10 children each from vaccinated and unvaccinated groups, among children who did not have any influenza vaccination, nor any virologically or serologically confirmed infection by influenza A/sH1N1, A/pH1N1, A/H3 and B viruses, in subsequent years of follow-up until Year 5, and tested their pre-/ post-vaccination and five post-epidemic sera. In sub study (iii), to study the protection against pH1N1 infection by vaccination, due to limited sample size, we selected all children with serologically confirmed pH1N1 infection in year 1 whether they were vaccinated or not, and randomly selected 10 children without serologically confirmed pH1N1 infection in year 1 from each of the three age groups in both vaccinated and unvaccinated groups, and tested their Pre inf (equivalent to Post vaxx) sera. Lastly, in sub study (iv), to study the diversity of antibody responses after a combination of vaccination and infection, we randomly selected 5 children from each of the three age groups in both vaccinated and unvaccinated groups, in both uninfected and infected children with serologically confirmed pH1N1 infection in year 1. Overall, we studied the vaccine-induced short-term cellular ADCC, long-term cellular ADCC, IgG, IgA and Fc γ R against multiple viral proteins, homologous (sH1N1) and cross-reactive (pH1N1) antibody response, cross-reactive response (IgG, IgA and Fc γ R) against infection by pH1N1, and breadth (IgG, IgM, IgA and Fc γ R) of response against multiple H1 and H3 virus strains by vaccination or infection (Table 2).”

To clarify, “in a separate set of participants (sub study ii)” has been added to the methods section.

Examples of organizational issues:

R4.3: Section beginning "Children who do not acquire pH1N1 infection have higher baseline pandemic-specific ADCC responses" deviates from this objective and contrasts vaccinated and unvaccinated participants, looks at age vs ADCC responses, and more.

A4.3: We have updated the title of this results section which attempted to describe the main result ‘take home message’

From: “Children who do not acquire pH1N1 infection have higher baseline pandemic-specific ADCC responses”,

To: “Features of antibody responses prospective to pandemic infection”

R4.4: Several results sections have discussion and limitations mixed in, which makes the results hard to follow.

Examples of lack of clarity in results: Lines 188-194 describing longitudinal data analysis is too vague to be of any use.

A4.4: We agree and apologise to the reviewer, this section has been revised for clarity to specifically describe results of interest:

“Significant fold change increases (versus theoretical value of 1) were observed shortly after vaccination (day 30, as 2009) for sH1-IgG, sH3-IgG, pH1-IgG, pH1-IgG1 and pH1-Fc γ R2a (Fig. 2b-l), but each returned to baseline within one year at 2010 (ns), and were not elevated compared to unvaccinated children. Other responses were analysed but were not significantly increased by vaccination (Extended data 1). Significant waning (by mixed effects model) of responses in unvaccinated (V0) children is evident across a number of antibody measures, between 2009 to later years for pH1-IgG (Fig. 2e), pH1-IgG1 (Fig. 2j), pH1-Fc γ R2a (Fig. 2l), pN1-IgG (Fig. 2g), H1stem-IgG (Fig. 2h), and M1-IgG (Fig. 2i).”

Examples of issues with statistical methods:

R4.5: Dunnett's test is appropriate for pairwise comparisons among groups following an overall result suggesting at least one difference within the set, but it is mentioned in the context of only two groups (eg vaccine vs placebo or pre- vs post-vaccination), so seems inappropriate.

A4.5: In Fig. 3 and 4, we use the Dunn's multiple comparisons test, following Kruskal Wallis test which revealed a significant difference between the four vaccine and infection outcomes (unvaccinated uninfected, unvaccinated infected, vaccinated uninfected and vaccinated infected). The test revealed which of the 4 outcomes had significant differences between each other. To directly compare pre- vs post- vaccination, we use Wilcoxon matched-pairs signed rank test in Fig. 1 and 2.

The methods section has been revised for clarification:

“Statistically significant differences in paired pre- versus post- vaccine responses were determined by determined by Wilcoxon matched-pairs signed rank test. For comparisons between vaccine groups and infection outcomes, a Kruskal Wallis test with Dunns multiple comparisons was used.”

R4.6: Correlates of risk tests seem to be done both using simple two-group comparisons (infected vs. uninfected vaccinees) as well as using logistic regression. Why the former is used at all is not clear, as the regression model is more powerful in that covariates and other immune markers can be adjusted.

A4.6: As our study set contains multiple groups and variables we have presented data progressively in various forms- initially two group comparisons were used in volcano plot analysis (Fig. 4ghi and Extended data 4abc) to highlight immune correlates between groups, whilst the logistic regression model (Extended data 5) is ultimately used to determine the contribution across 2 sample sets of the same shared variables. We agree with the reviewer that 2 group comparison are not as powerful but for clarity have retained these figures as they help distil important immune features.

R4.7: The logistic models fit are massively overparameterized: one model has 24 variables and is fit using 115 observations. A general rule of thumb is no fewer than 10 observations per parameter, which would mean that 250 observations would be needed to fit such a model.

A4.7: We are grateful for the reviewer's insightful comment on the overparameterization of the model. Following the suggestion, we have revised our model to include only the top 10 most important variables out of the original 24. In addition, the logistic regression model were built with stepwise variable selection by AIC, which also helps reduce the fitting variables in the final model. As a results, our final Model 1 and 2 use 7 and 4 variables respectively.

The 10 input variables were identified based on the Elastic Net and PLSDA feature selection results (Extended data 5d) for sub-studies (iii) and (iv). Details explaining the rationale for choosing the variables in the model are added to the Methods section (Lines 656-666). We have also revised the main text (Lines 312-342) and the figures (Extended data 5 a,b,c) accordingly. Generally, the model results are similar to the previous model with 24 variables. The most significant variables for model 1 (IgG1_sH3_2007 and IgG3_sH3_2007) and model

2 (pre_infection_HAI_pH1 and IgG1_sH3-2007) remain the same, except for the order of the variables. The accuracy of the models for fitting and testing data slightly decreased as expected (fewer variables included and thus prone to underfit), but the conclusion is not affected. We still find that the performance of prediction is not high, and input of more shared antibody features may improve the model. (Lines 338-342)

The main text and extended data 5 have been updated.

Extended data 5 (a-c) are updated, but (d) remains the same. In (d), labelled in bold, the 10 variables we selected and input into model 1 and 2, are:

pre_infection_HAI_pH1
IgG1_sH3-2007
IgG3_sH3-2007
IgG1_H1stem
IgG1_vaxx_sH1-2007
IgA1_pH1-2009
IgG1_pH1-2009
IgG3_H1stem
FcyR3a_pH1-2009
FcyR3a_vaxx_sH1-2007

R4.8: Removing outliers is generally not good practice, unless there is strong evidence that there are actual data entry or assay failure; observing unusually high or low values don't provide sufficient rationale for exclusion

A4.8: We have edited the longitudinal graphs and statistics in Fig. 2 to include outliers. This has not impacted results.

Reviewer #4 (Remarks to the Author):

The authors have adequately responded to the reviewers' comments.